# Treatment outcomes of multi drug resistant and rifampicin resistant Tuberculosis in Zimbabwe: A cohort analysis of patients initiated on treatment during 2010 to 2015

**Ronnie Matambo**[1]*, **Kudakwashe C. Takarinda**[2,3], **Pruthu Thekkur**[2,4], **Charles Sandy**[3], **Sungano Mharakurwa**[5], **Talent Makoni**[3], **Ronald Ncube**[1], **Kelvin Charambira**[1], **Christopher Zishiri**[1], **Mkhokheli Ngwenya**[6], **Saziso Nyathi**[7], **Albert Chiteka**[5], **Elliot Chikaka**[5], **Shungu Mutero-Munyati**[8]

**1** International Union against Tuberculosis & Lung Disease, Harare, Zimbabwe, **2** Centre for Operational Research, International Union against Tuberculosis & Lung Disease, Paris, France, **3** AIDS and TB Department, Ministry of Health & Child Care, Harare, Zimbabwe, **4** The Union South East Asia (The USEA) Office, New Delhi, India, **5** College of Health, Agriculture and Natural Sciences, Africa University, Mutare, Zimbabwe, **6** World Health Organisation, Zimbabwe Country Office, Harare, Zimbabwe, **7** Health Services Department, City of Bulawayo, Zimbabwe, **8** Biomedical Research and Training Institute, Harare, Zimbabwe

* rmatambo2@gmail.com

**Data Availability Statement:** All data generated from the National TB Programme of Zimbabwe are under ownership of the Government of Zimbabwe

## Abstract

### Background

Zimbabwe is one of the thirty countries globally with a high burden of multidrug-resistant tuberculosis (TB) or rifampicin-resistant TB (MDR/RR-TB). Since 2010, patients diagnosed with MDR/RR-TB are being treated with 20–24 months of standardized second-line drugs (SLDs). The profile, management and factors associated with unfavourable treatment outcomes of MDR/RR TB have not been systematically evaluated in Zimbabwe.

### Objective

To assess treatment outcomes and factors associated with unfavourable outcomes among MDR/RR-TB patients registered and treated under the National Tuberculosis Programme in all the district hospitals and urban healthcare facilities in Zimbabwe between January 2010 and December 2015.

### Methods

A cohort study using routinely collected programme data. The 'death', 'loss to follow-up' (LTFU), 'failure' and 'not evaluated' were considered as "unfavourable outcome". A generalized linear model with a log-link and binomial distribution or a Poisson distribution with robust error variances were used to assess factors associated with "unfavourable outcome". The unadjusted and adjusted relative risks were calculated as a measure of association. A $p$ value< 0.05 was considered statistically significant.

through the Ministry of Health and Child care. Data can only be obtained by extending your request to the Permanent Secretary for Health through the National TB Programme Manager and with concurrence from the local ethics board. Request for data sharing may be directed to the following: 1. The Chairperson, Medical Research Council of Zimbabwe (MRCZ), Josiah Tongogara/Mazowe Street, Box CY 573, Causeway, Harare (Email: mrcz@mrcz.org.zw). 2. Dr Charles Sandy, National TB Programme Manager, Ministry of health and child care, P.O box cy1122, Causeway, Zimbabwe (Email: dr.c.sandy@gmail.com). To request the data, please refer to the MRCZ project number (MRCZ/A/2331) and the dataset name (R_Matambo_MDR_TB_study_dataset.dta).

**Funding:** The training course under which this study was conducted was funded by: the United Kingdom's Department for International Development (DFID); The Global Fund to Fight AIDS, Tuberculosis and Malaria (GFATM) and the World Health Organization. The open access publications costs were funded by the Department for International Development (DFID), UK and La Fondation Veuve Emile Metz-Tesch (Luxembourg). The funders had no role in study design, data collection and analysis, decision to publish, or preparation of the manuscript.

**Competing interests:** The authors have declared that no competing interests exist

## Results

Of the 473 patients in the study, the median age was 34 years [interquartile range, 29–42] and 230 (49%) were males. There were 352 (74%) patients co-infected with HIV, of whom 321 (91%) were on antiretroviral therapy (ART). Severe adverse events (SAEs) were recorded in 118 (25%) patients; mostly hearing impairments (70%) and psychosis (11%). Overall, 184 (39%) patients had 'unfavourable' treatment outcomes [125 (26%) were deaths, 39 (8%) were lost to follow-up, 4 (<1%) were failures and 16 (3%) not evaluated]. Being co-infected with HIV but not on ART [adjusted relative risk (aRR) = 2.60; 95% CI: 1.33–5.09] was independently associated with unfavourable treatment outcomes.

## Conclusion

The high unfavourable treatment outcomes among MDR/RR-TB patients on standardized SLDs were coupled with a high occurrence of SAEs in this predominantly HIV co-infected cohort. Switching to individualized all oral shorter treatment regimens should be considered to limit SAEs and improve treatment outcomes. Improving the ART uptake and timeliness of ART initiation can reduce unfavourable outcomes.

## Introduction

Globally, Tuberculosis (TB) remains a major public health concern with an estimated 10 million incident TB patients and 1.6 million deaths due to the disease in 2017.[1] In recent years, drug-resistant TB (DR-TB) has emerged globally. Drug-resistant TB is posing an additional threat to TB control efforts due to the complexity of its treatment and poor treatment outcomes. The World Health Organization (WHO) estimated that there were about 558,000 incident multidrug-resistant TB or rifampicin-resistant TB (MDR/RR-TB) patients globally in the year 2017, but only 29% were notified. The WHO also estimated that 3.5% of new TB cases and 18% of previously treated TB patients had MDR/RR-TB in 2017.[1]

In 2000, WHO recommended the 20–24 month standardized second-line drug (SLDs) regimens for the treatment of MDR/RR-TB patients in resource-limited settings.[2] The WHO also set a target of achieving a 75–90% treatment success rate (i.e., cured or treatment completed) by 2015.[3] However, the 2018 global TB report shows that only 55% of patients initiated on MDR/RR-TB treatment during the year 2015 had successful treatment outcomes.[1] A recent systematic review in 2017 reported successful treatment outcomes among MDR/RR-TB patients on standardized SLD regimen to be 64%. The studies included in the review were mostly from countries with low HIV coinfection and thus, recommended for systematic assessment of treatment outcomes in high HIV coinfection countries of sub-Saharan Africa. [4]

Treatment of MDR/RR-TB is not only longer, more complex and expensive but also involves the use of drug regimens that are more toxic and could lead to severe side effects, such as deafness and liver damage.[5,6] Though, recently WHO recommended shorter regimens for management of MDR/RR-TB[14], low-middle income countries still largely use longer standardized SLD regimens requiring patients to consume drugs for not less than eighteen months.

Studies conducted in various countries suggest that MDR/RR-TB treatment outcomes vary by factors specific to individual patients and are also related to TB program implementation. [7–10] Patient-level characteristics like HIV-coinfection, alcohol and substance use, smoking

and low body mass index have been found to be associated with unsuccessful treatment outcomes. Programmatic characteristics like delay in treatment initiation, duration of treatment, type of drug sensitivity testing, individualized treatment regimens and use of directly observed therapy have also been found to be associated with adverse MDR/RR-TB treatment outcomes. [7–10]

Zimbabwe, which is located in southern Africa, is among the 14 high burden countries (HBCs) with a triple burden of TB, TB/HIV and MDR-TB.[1] In Zimbabwe, there were an estimated 37,000 incident TB patients and 8,300 TB-associated deaths in 2017.[1] In the same year, there were an estimated 1,300 MDR/RR-TB patients in the country with a prevalence of 4.6% and 14% among new and previously treated TB patients, respectively.[1] In 2010, the National TB Programme (NTP) of Zimbabwe released the "Programmatic Management of Drug-Resistant TB" (PMDT) guidelines with the use of standardized SLDs for twenty months. [11] In 2016, the 9–11 month injectable-based short treatment regimen was adopted as a pilot but in only one of the country's districts, which was supported by a non-governmental organisation. This has been however rolled out by the NTP in some of the districts in Zimbabwe

In 2017, WHO estimated that less than 40% of the MDR/RR-TB patients were diagnosed and put on treatment in Zimbabwe.[12] Also, among those initiated on treatment, more than 50% had unsuccessful treatment outcomes. High unsuccessful treatment outcomes in Zimbabwe are largely due to the fact that treatment outcomes of about 32% are missed during routine reporting and are eventually considered as 'unsuccessful' treatment outcomes.[12] There has been no systematic assessment of treatment outcomes of MDR/RR-TB patients treated under the Zimbabwe NTP, nor has there been assessment of the individual and programmatic characteristics associated with unsuccessful treatment outcomes. Knowledge of factors affecting outcomes among MDR/RR-TB patients can guide the NTP to make informed decisions on policies and strategies aimed at improving treatment outcomes for subsequent MDR/RR-TB patient cohorts. We therefore conducted a study aimed at assessing the profile, treatment outcomes and factors associated with unfavourable treatment outcomes among patients initiated on MDR/RR-TB treatment under the Zimbabwe NTP between 2010 and 2015.

## Methods

### Study design

This was a cohort study using secondary data routinely collected from fifty-two out of sixty-three district hospitals and thirty out of thirty-five Urban Poly-clinics in the two metropolitan provinces within the Zimbabwe NTP.

### General setting

Zimbabwe is a landlocked country with an estimated population of 17 million.[1] The country has 62 districts, which are further grouped into ten provinces of which two are Metropolitan provinces (Harare, the capital city and Bulawayo, the second-largest city). The country's public healthcare referral system constitutes four levels: 1) the quaternary level constituting six central hospitals located in the two metropolitan provinces 2) the tertiary level consisting eight provincial hospitals which are the highest referral hospitals providing selected basic medical specialties for the eight rural provinces 3) the secondary level constituting at least one district and or general hospital per district and last 4) the primary care level consisting of rural and urban healthcare facilities that provide primary health care services.

**Specific setting.** *Diagnosis of MDR/RR-TB*. In Zimbabwe, TB diagnosis and treatment services are provided in public healthcare facilities and are integrated with general health services. Private laboratories complement efforts of public healthcare by providing TB diagnosis

services for patients seeking care in private sector. Before 2013, only previously-treated sputum positive pulmonary tuberculosis patients (PTB) and MDR-TB contacts were considered as presumptive MDR-TB patients and evaluated for MDR-TB. Their sputum specimens were subjected to phenotypic culture and drug susceptibility testing (CDST) with BBL™ MGIT™ Mycobacterial Growth Indicator Tubes Becton Dickinson, Sparks, MD or genotypic Micobacterium Tuberculosis Rifampicin-resistant (MTB/Rif) assay. From 2013 onwards, Xpert MTB/Rif assay was used upfront for diagnosis of TB and rifampicin resistance in MDR-TB high-risk groups (retreatment TB patients, chest symptomatic MDR-TB contacts, those HIV-positive, health workers with pulmonary TB miners with PTB and children <5 years). In all RR-TB patients, the remainder of the two collected sputum specimens is sent to one of the country's two national reference laboratories for CDST in order to assess drug susceptibility to all the first-line drugs.[11]

*Treatment initiation and follow-up MDR/RR-TB.* All diagnosed MDR/RR-TB patients are registered and started on treatment at either district or provincial hospitals or at polyclinics and infectious disease hospitals in metropolitan provinces. The District Medical Officer is responsible for providing oversight on the clinical management of all MDR/RR-TB patients in their respective districts.

On registration at the district hospital, the patient is notified to the NTP and a patient-held DR-TB treatment card is issued. Simultaneously, the socio-demographic and clinical details of the patient are documented in the Daily Observed Treatment (DOT) DR-TB register maintained at health facility. The patient follow-up visit details are updated regularly in the 'DOT DR-TB' by the healthcare workers. Patient data in the health facility 'DOT DR-TB' register are also entered into the DR-TB register which is maintained and updated by DR-TB co-ordinators at district level.

As part of pre-treatment evaluation for all patients, laboratory investigations like liver function tests, renal function tests and complete blood count are supposed to be done before initiation of treatment. The district clinical management team initiates treatment based on pre-treatment evaluation. Details of the clinical and laboratory examinations are documented in the DR-TB card and also in the clinical notes attached to the DR-TB card. During the study period, the WHO recommended use of a standardised DOTS-Plus regimen for the management of MDR/RR-TB patients.[13,14] The duration of treatment was at least 20 months with a minimum of six months (and four months after culture conversion) in the intensive phase and 14 months of the continuation phase. Oral drugs, namely levofloxacin, pyrazinamide, cycloserine and ethambutol were given both during the intensive and continuation phases. The injectable kanamycin was provided six days a week during the intensive phase. Treatment dosages were dispensed based on patient weight.

After treatment initiation, patients are monitored for two weeks at facilities where treatment is initiated before considering if the patient is stable and tolerating the regimen. Those considered "Stable" were patients who were able to ingest medication, did not show signs of adverse drug reaction and had all the laboratory investigations within normal limits. Based on clinical severity and distance of travel from the patient's residence (>10 km), patients are either admitted and monitored or asked to visit the district hospital daily. After two weeks, based on proximity to a health facility, patients either continue DOTS-Plus in district hospitals or they are referred to primary health facilities nearest to their residence. Patients are followed-up as per PMDT guidelines and the programmatic treatment outcomes are ascertained by the medical officer (**Table 1**). If a patient developed side-effects due to kanamycin, their dosage was reduced; however, currently, they are switched to capreomycin.[11]

Patients are also offered provider-initiated HIV testing services in the MDR-TB pre-treatment phase and those found to be HIV positive are assessed for initiation on antiretroviral

**Table 1. Treatment outcome definitions.**

| TREATMENT OUTCOME | Definition |
|---|---|
| Treatment completed | -A patient who has completed treatment but who does not have a negative sputum smear or culture result in the last month of treatment and on at least one previous occasion. The sputum-smear or culture may not have been done or the results are not available. |
| Cure | A patient whose sputum smear or culture was positive at the beginning of treatment and who was sputum smear or culture-negative in the last month of treatment and on at least one previous occasion. |
| Treatment success | The sum of patients who were initially sputum smear or culture positive and were cured and those who completed treatment. |
| Treatment Failure | Patient who is sputum smear or culture-positive at 5 months or later during treatment. Patient who was initially smear-negative before starting treatment and became smear-positive after completing the intensive phase of treatment. Any patient who is found to have MDR-TB at any point of time during the treatment, whether they are smear negative or -positive |
| Died | Patient who dies for any reason during the course of treatment |
| Default/ Lost to Follow-up | Patient whose treatment was interrupted for two consecutive months or more |
| Transfer out | Patient who has been transferred to another recording and reporting unit and for whom the treatment outcome is not Known. |
| Not evaluated | Patient whose treatment outcomes are not ascertained |

-

Source: Ministry of Health and Child Care. Guidelines for the Programmatic Management of Drug Resistant Tuberculosis in Zimbabwe. Harare, Zimbabwe; 2014.

therapy (ART).and also Cotrimoxazole preventive therapy (CPT) as per WHO guidelines. As per the national guidelines in use during the study period, they were initiated on a fixed-dose combination once-daily pill of Tenofovir v plus(+) Lamuvidine (or Emtricitabine) plus (+) Efavirenz (TDF+3TC (or FTC)+EFV) as the preferred first-line ART regimen among adult PLHIV. In children living with HIV abacavir+lamuvidine+efavirenz (or Lopinavir/Ritonavir) (ABC + 3TC + EFV (or LPV/r)) was the preferred first-line ART regimen.[15]

**Study population.** All MDR/RR-TB patients initiated on treatment between 2010 and 2015 under the Zimbabwe NTP and continued their treatment at either district hospitals or urban polyclinics were included in the study. Those patients who were referred back to primary health facilities for DOTS-Plus treatment were excluded due to resource and time constraints in travelling to all primary healthcare facilities to collect their socio-demographic and clinical details.

**Data variables, sources of data and data collection.** Patient demographic and clinical data were extracted from the health facility DOT register, individual patient clinical notes and the district DR-TB register using a structured proforma. Data extraction was done by District TB coordinators of the respective districts following training by the principal investigator. A data extraction manual was also shared by the principal investigator indicating the source of variables and explaining the standard procedure to be followed while extracting each variable. District Environmental Health Officers (DEHOs) of the respective districts also crosschecked the source registers and validated 10% of the extracted data. Data was extracted from August to December 2018.

*Operational definitions*:

*Duration from diagnosis to treatment initiation*: The number of days between the diagnosis date of rifampicin resistance to date of initiating standardised SLDs for management of MDR/RR-TB.

*Severe Adverse Events (SAEs)*: All the adverse events as listed in PMDT guidelines of Zimbabwe.[11]

*Other comorbidities*: All the self-reported comorbidities (except for HIV) recorded during the initiation of treatment.

## Data entry, analysis and reporting

Data were double entered and validated using EpiData Entry software (EpiData Association, Odense, Denmark). Data were analysed using EpiData analysis (version 2.2.2.182, EpiData Association, Odense, Denmark) and Stata (version 12.0 STATA Corp., College, TX, USA). Study findings were reported in accordance with Strengthening the Reporting of Observational Studies in Epidemiology (STROBE) guidelines.[16]

Categorical variables such as MDR/RR-TB treatment outcomes were summarized using numbers and percentages while medians (interquartile range (IQR)) were calculated for skewed continuous data such as age and weight at treatment initiation.

The TB treatment outcomes were ascertained at completion of treatment unless there was a known outcome such as deaths, LTFUs or treatment failure. Those classified as LTFU were defined as patients whose MDR-TB treatment was interrupted for two or more consecutive months for any reason. All treatment outcomes were defined in line with WHO guidelines. The primary outcome, "unfavorable outcome" was comprised of 'death', 'loss to follow-up' (LTFU), 'failure' and 'not evaluated' while 'cured' and 'treatment completed' comprised 'favorable outcome' in line with recent meta-analysis and systematic reviews on MDR-TB treatment outcomes.[4,9]

To assess factors associated with "unfavorable outcome", we used univariate and multivariate generalized linear model with a log-link and binomial distribution or alternatively a Poisson distribution with robust error variances, as the binomial model failed to converge. All potential factors with a $p \leq 0.25$ in the unadjusted model were included in the multivariate regression model. Unadjusted and multivariate-adjusted relative risks with 95% confidence interval (CI) were calculated as a measure of association. A *p* value$< 0.05$ was considered statistically significant.

## Ethics approval

Ethics approval was granted by The Union Ethics Advisory Group of the International Union against Tuberculosis and Lung Diseases, Paris, France, IRB number EAG 53/18 and the Medical Research Council of Zimbabwe (MRCZ), IRB number MRCZ/A/2331 Permission to access data was granted from the Ministry of Health and Child Care. No patient consent was required as this was already granted by the Ministry of health and child care on behalf of the patients since this was a retrospective study. Data were fully anonymised as only DR-TB registration numbers were abstracted onto data collection proformas as unique patient identifiers. The Ethics committee waived the requirement for individual informed consent.

## Results

Of the total 935 MDR/RR-TB patients initiated on treated during the study reference period, 473 (51%) were followed up at district hospitals and urban healthcare facilities and were included in the study. The 473 patients in our study contributed 672.5 person-years of follow-up time. The median age of these 473 participants was 34 (IQR, 29–42) years and 230 (49%) were males. The demographic details of the participants are shown in **Table 2**.

**Table 3** shows the clinical characteristics of the study participants. Of all patients, 257 (54%) were new TB patients, followed by 116 (25%) with 'retreatment after failure'. Of the 352

**Table 2. Baseline demographic characteristics of MDR/RR-TB patients initiated on treatment during 2010 to 2015 in Zimbabwe.**

| Characteristic | N (%) |
|---|---|
| **Total** | 473 (100) |
| Sex | |
| Male | 230 (48.6) |
| Female | 241 (51.0) |
| Missing | 2 (<1) |
| Age | |
| <5 | 2 (<1) |
| 5–14 | 10 (2.1) |
| 15–24 | 56 (11.8) |
| 25–34 | 169 (35.7) |
| 35–44 | 149 (31.5) |
| 45–54 | 47 (9.9) |
| 55+ | 34 (7.2) |
| Not recorded | 6 (1.3) |
| *Median (IQR)* | *34 (29–42)* |
| Marital status | |
| Married | 202 (42.7) |
| Single | 143 (30.2) |
| Widowed | 44 (9.3) |
| Divorced | 25 (5.3) |
| Missing | 59 (12.5) |

MDR-TB = multidrug resistant tuberculosis; RR-TB = rifampicin resistant tuberculosis; IQR = interquartile range

(74%) study participants co-infected with HIV, 321 (91%) were on ART and 323 (91%) received CPT. The majority of the HIV-positive and ART naïve, and with unknown HIV status were new TB patients. Of the 473 participants, 402 (85%) were diagnosed with a genotypic test and 290 (68%) were initiated on MDR-TB treatment within seven days of MDR/RR-TB diagnosis while it was more than four weeks in 64 (14%) of the participants. Overall, the median duration from diagnosis to initiation of MDR/RR-TB treatment was 1 (IQR, 0–13) days.

Overall, 17 (4%) patients had a documented comorbidity other than HIV. These 17 had different comorbidities with the most common being, 3/17 (18%) anaemia, 2/17 (12%) diabetes mellitus, and 2/17 (12%) renal failure. Among all the participants, 118 (25%) developed SAEs during treatment. Among those with SAEs, 80 (68%) had hearing disorders/impairments, 14 (12%) had psychosis, 9 (8%) had jaundice,8 (7%) had Steven Johnson syndrome and another 8 (7%) had nausea, diarrhoea and vomiting (**Table 4**).

The data on culture conversion and MDR-TB treatment outcomes among study participants are shown in **Table 5**. Of the 287 (61%) participants with recorded culture conversion results, 259 (90%) had culture conversion within six months of treatment initiation. Of all patients, 184 (39%), (95% CI: 34.5–44.5) patients had an 'unfavourable' treatment outcome of whom 125 (26%) died while 39 (8%) were lost to follow-up. The proportion with an unfavourable treatment outcome (death, Treatment failure, LTFU and Not evaluated) increased annually from 0% in 2010 to 45% in 2015 (**Fig 1**).

Factors associated with an unfavourable outcome among patients on MDR-TB treatment are shown in **Table 6**. Those who were HIV-positive and ART-naive (ARR = 2.60; 95% CI: 1.33–5.09) were more likely to have an unfavourable treatment outcome

**Table 3. Clinical characteristics of MDR/RR-TB patients at baseline and/or during MDR-TB initiated on treatment in Zimbabwe, 2010–2015.**

| Characteristic | | N | (%) |
|---|---|---|---|
| **Total** | | **473** | **(100)** |
| *Type of TB* | New | 257 | (54.3) |
| | Retreatment After LTFU | 19 | (4.0) |
| | Retreatment After Failure | 116 | (24.5) |
| | Retreatment Relapse | 81 | (17.1) |
| *Duration from diagnosis to treatment initiation* | ≤7 days | 290 | (61.3) |
| | 8–30 days | 71 | (15.0) |
| | 31–90 days | 32 | (6.8) |
| | >90 days | 32 | (6.8) |
| | Not recorded | 48 | (10.2) |
| *Type of TB diagnostic test* | Genotypic | 402 | (85.0) |
| | Phenotypic | 53 | (11.2) |
| *Baseline CDST of Isoniazid* | Sensitive | 35 | (7.4) |
| | Resistant | 172 | (36.4) |
| | Not recorded | 266 | (56.2) |
| *Baseline CDST of Ethambutol* | Sensitive | 104 | (22.0) |
| | Resistant | 65 | (13.7) |
| | Not recorded | 304 | (64.3) |
| *Baseline CDST of Streptomycin* | Sensitive | 99 | (20.9) |
| | Resistant | 66 | (14.0) |
| | Not recorded | 308 | (65.1) |
| *HIV status* | HIV negative | 101 | (21.4) |
| | HIV-positive | 352 | (74.4) |
| | Untested/unknown | 20 | (4.3) |
| *ART status (n = 352)* | Yes | 321 | (91.2) |
| | No | 18 | (5.1) |
| | Not recorded | 13 | (3.7) |
| *CPT status (n = 352)* | Yes | 323 | (91.8) |
| | No | 11 | (3.1) |
| | Not recorded | 18 | (5.1) |
| *Encountered SAE during treatment* | Yes | 118 | (25) |
| | No | 324 | (68.5) |
| | Not recorded | 36 | (7.6) |
| *Other comorbidities* | Yes | 17 | (3.6) |
| | No | 456 | (96.4) |

TB = Tuberculosis; MDR-TB = Multi drug Resistant TB; RR-TB = Rifampicin Resistant TB; DST = Drug sensitivity pattern; LTFU = Loss to follow-up; IQR = interquartile range; ART = Anti-retroviral treatment; CPT = cotrimoxazole prophylactic treatment; SAE = Serious Adverse events;

## Discussion

This was the first nationwide study in Zimbabwe to assess the profile, management and factors associated with unfavourable treatment outcomes of MDR/RR-TB patients. The key findings of the study, which are programmatically important, are listed here. 1) About three-quarters of the patients were co-infected with HIV; of them, one out of ten was not on ART. 2) About one-quarter of the patients developed SAEs and hearing impairment was the most common

**Table 4. Distribution and type of severe adverse events encountered during MDR-TB treatment among MDR/RR-TB patients in Zimbabwe, 2010–2015.**

| Severe Adverse Events | N | n | (%) |
|---|---|---|---|
| hearing disorder/impairment | 118 | 80 | (67.8) |
| psychosis | 118 | 14 | (11.9) |
| jaundice | 118 | 9 | (7.6) |
| Steven Johnson Syndrome | 118 | 8 | (6.8) |
| nausea, diarrhoea & vomitting | 118 | 8 | (6.8) |
| seizures | 118 | 4 | (3.4) |
| peripheral nueropathy | 118 | 3 | (2.5) |
| severe chest pains | 118 | 1 | (0.8) |
| blurred vision | 118 | 1 | (0.8) |
| Not recorded | 118 | 1 | (0.8) |

*Patients may have presented with ≥1 severe adverse events hence percentages
do not add up to 100%.

SAE. 3) Around six out of ten patients had favourable treatment outcomes and about one-quarter of the patients died during treatment. 4) One in seven patients encountered MDR-TB treatment delays following RR-TB diagnosis 5) Being co-infected with HIV but not being on ART was associated with having an unfavourable outcome.

This study had some strengths. First, this study included patients from all the districts spanning over five years since the country adopted standardized SLDs hence, the findings provide useful insights for follow-up programmatic decision making at national level. Second, the

**Table 5. Programmatic treatment outcomes of MDR/RR-TB patients initiated on treatment during 2010 to 2015 in Zimbabwe.**

| Treatment outcomes | N (%) |
|---|---|
| Total | 473(100) |
| **Culture conversion period** | |
| ≤ 6 months | 259 (54.8) |
| >6 months | 28 (5.9) |
| Not recorded | 186 (39.3) |
| **End of TB treatment outcomes** | |
| *Favourable outcomes*[a] | 289 (61.1) |
| Treatment completed | 149 (31.5) |
| Cured | 140 (29.6) |
| *Unfavourable outcomes*[b] | 184 (38.9) |
| Died | 125 (26.4) |
| LTFU | 39 (8.2) |
| Failed | 4 (0.8) |
| Not evaluated | 16 (3.4) |

TB = Tuberculosis; MDR-TB = Multi drug Resistant TB; RR-TB = Rifampicin Resistant TB;

LTFU = loss to follow up

a–favourable outcomes is a combination of those who completed treatment and who were cured

b–adverse outcomes is a combination of those who died, where LTFU, failures and not evaluated

NB: Please note that the end of TB treatment outcomes do not add up to 100% because of rounding-off errors

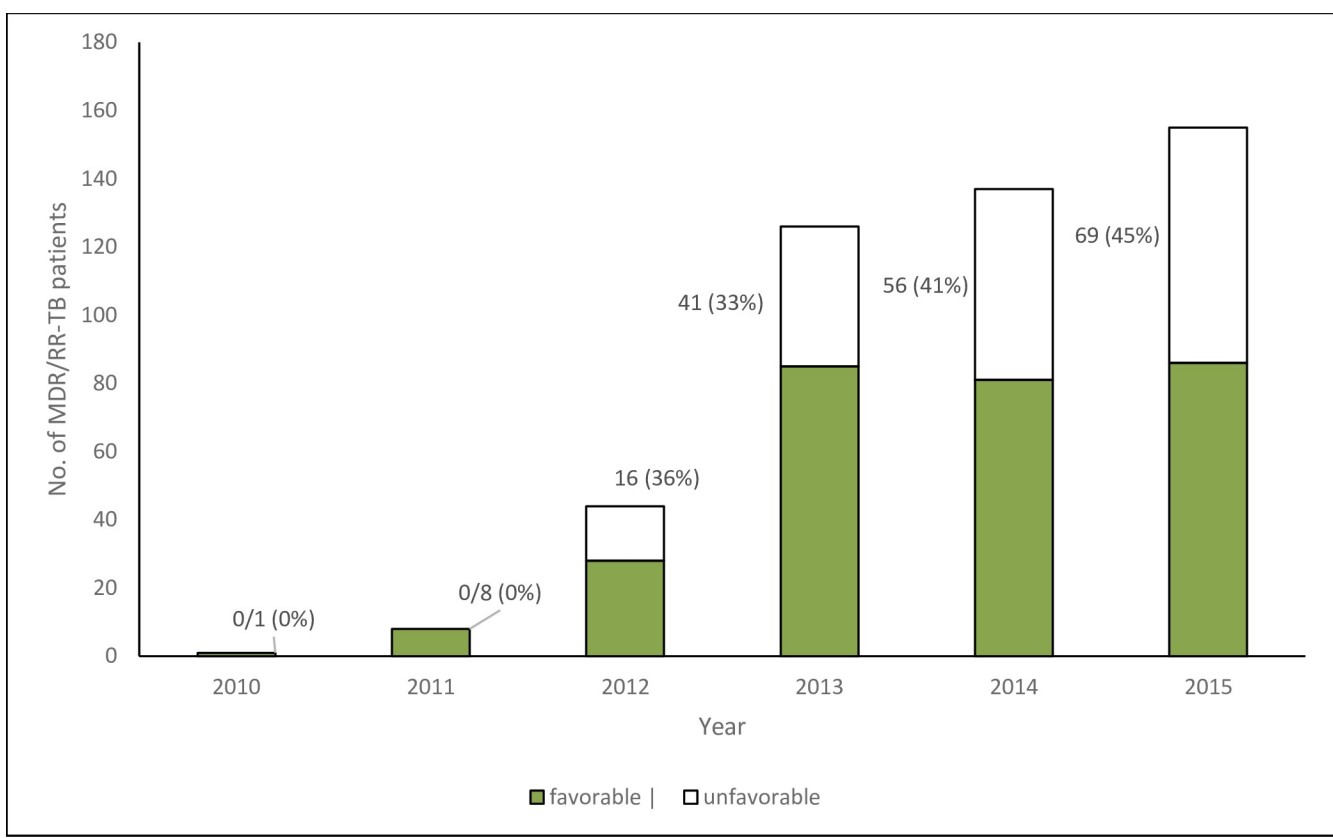

**Fig 1. Treatment outcomes among MDR/RR TB patients registered in Zimbabwe between 2010 and 2015, stratified by year of treatment commencement.**

study was conducted under a routine programmatic setting, thus highlighting important operational challenges that are applicable to other NTPs in low-resource settings.

The study was not without limitations. First, the patients who were referred back to rural primary healthcare facilities for follow-up care after MDR-TB treatment initiation were not included. Compared to patients referred back, the patients included in our study were more likely to be from urban areas with better socio-economic status, education levels and access to healthcare services. Thus, the current study cohort is more likely to have had better treatment outcomes and higher treatment success rates. Second, there were missing data on key variables which include CDST results, socio-economic status, WHO clinical staging, CD4 cell count, timing of ART in relation to MDR-TB treatment commencement, nutritional status, MDR-TB drug regimens and their dosages, data on virological, immunological or clinical failure of patients initiated on ART and radiological findings of TB lesions–all which are important factors which may be related to MDR-TB treatment outcomes. Third, while comorbidities were recorded, these were determined during eliciting patient history at MDR-TB treatment initiation and were self-reported by patients, hence, there might have been an underestimation of prevailing comorbidities like diabetes mellitus, which require specific diagnostic tests. Fourth, details on management of SAEs such as changes in drug regimen or dosages and their effect on treatment outcomes were not assessed.

Though the overall treatment success rate from our study was lower than the 75–90% target recommended by WHO, this was higher than the 52% success rate with standardized regimens as reported in a recent meta-analysis.[17] The treatment success rate was also better than that

**Table 6. Sociodemographic and clinical characteristics associated with unfavourable outcomes among MDR/RR-TB patients initiated on treatment during 2010 to 2015 in Zimbabwe.**

| Characteristic | No. on RR/MDR-TB treatment | Unfavourable Outcome | RR (95% CI) | ARR (95% CI) |
|---|---|---|---|---|
| **Total** | **473** | **184 (38.9)** | | |
| Sex | | | | |
| Male | 230 | 92 (40.0) | Reference | Reference |
| Female | 241 | 92 (38.2) | 0.95 (0.76–1.20) | 1.01 (0.75–1.37) |
| Not recorded | 2 | 0 (0.0) | - | - |
| Age | | | | |
| <24 | 68 | 21 (30.9) | Reference | Reference |
| 25–34 | 169 | 61 (36.1) | 0.86 (0.57–1.29) | 0.85 (0.51–1.43) |
| 35–44 | 149 | 57 (38.3) | 1.06 (0.80–1.41) | 1.03 (0.71–1.49) |
| 45–54+ | 47 | 24 (51.1) | 1.41 (1.00–2.00) | 1.35 (0.83–2.19) |
| 55+ | 34 | 19 (55.9) | 1.55 (1.08–2.22) | 1.41 (0.83–2.4) |
| Not recorded | 6 | 2 (33.3) | 0.92 (0.29–2.91) | 0.66 (0.15–2.78) |
| Time from RR-TB diagnosis to MDR-TB treatment initiation | | | | |
| ≤7 days | 290 | 125 (43.1) | Reference | Reference |
| 8–30 days | 71 | 19 (26.8) | 0.62 (0.41–0.93) | 0.64 (0.39–1.06) |
| 31–90 days | 32 | 11 (34.4) | 0.80 (0.49–1.31) | 0.91 (0.48–1.72) |
| >90 days | 32 | 11 (34.4) | 0.80 (0.49–1.31) | 0.89 (0.47–1.67) |
| Not recorded | 48 | 18 (37.5) | 0.87 (0.59–1.28) | 0.97 (0.56–1.66) |
| Isoniazid DST pattern | | | | |
| Sensitive | 35 | 5 (14.3) | Reference | Reference |
| Resistant | 172 | 54 (31.4) | 2.20 (0.95–5.10) | 2.12 (0.82–5.45) |
| Not recorded | 266 | 125 (47.0) | 3.29 (1.45–7.48) | 2.69 (0.97–7.51) |
| Ethambutol DST pattern | | | | |
| Sensitive | 104 | 32 (30.8) | Reference | Reference |
| Resistant | 65 | 16 (24.6) | 0.80 (0.48–1.34) | 1.06 (0.52–2.16) |
| Not recorded | 304 | 136 (44.7) | 1.45 (1.06–1.99) | 0.85 (0.33–2.15) |
| Streptomycin DST pattern | | | | |
| Sensitive | 99 | 30 (30.3) | Reference | Reference |
| Resistant | 66 | 14 (21.2) | 0.70 (0.40–1.22) | 0.74 (0.35–1.54) |
| Not recorded | 308 | 140 (45.5) | 1.50 (1.09–2.07) | 1.33 (0.56–3.14) |
| HIV & ART status | | | | |
| HIV-ve | 101 | 27 (26.7) | Reference | Reference |
| HIV+ve, on ART | 321 | 126 (39.3) | 1.47 (1.03–2.08) | 1.37 (0.89–2.11) |
| HIV+ve, not on ART | 18 | 14 (77.8) | 2.91 (1.94–4.37) | 2.6 (1.33–5.09) |
| HIV+ve, ART unknown | 13 | 8 (61.5) | 2.3 (1.34–3.94) | 2.07 (0.9–4.78) |
| HIV status unknown | 20 | 9 (45.0) | 1.68 (0.94–3.01) | 1.87 (0.85–4.14) |
| CPT status | | | | |
| Yes | 323 | 133 (41.2) | Reference | Reference |
| No | 11 | 6 (54.6) | 0.82 (0.51–1.33) | - |
| Not recorded | 18 | 9 (50.0) | 1.09 (0.54–2.22) | - |
| Other comorbidities recorded | | | | |
| Yes | 17 | 10 (58.8) | 0.65 (0.43–0.98) | 0.64 (0.33–1.24) |
| No | 456 | 174 (38.2) | Reference | Reference |

TB = Tuberculosis; MDR-TB = multi-drug resistant TB; RR-TB = rifampicin resistant TB; RR = relative risk; ARR = multivariate-adjusted relative risk; DST = drug sensitivity testing; ART = antiretroviral therapy; CPT = cotrimoxazole preventive therapy; SAE = severe adverse event

reported in similar patient cohorts from South Africa[18,19] and Zambia[20], where the majority of patients were co-infected with HIV. However, the better treatment outcomes in the current study may be due to potential selection bias of excluding patients treated at primary health centres. In contrast, better treatment success rates of around 75% have been observed in similar cohorts in Botswana and eSwatini with similar high HIV and MDR/RR-TB co-infection MDR/RR-TB cohorts.[21,22] However, the patient in Botswana and eSwatini had received social support in the form of monetary incentives or nutritional and transportation support including psychological support at health facilities. No such social and psychological support mechanisms were made available in the current study setting and should be considered in the future for improving treatment outcomes.

Similar to previous studies, patients co-infected with HIV but not on ART had a higher risk of having unfavourable outcomes in particular death when compared to those who are HIV-negative.[23,24] It is likely that such ART-naïve HIV co-infected patients were presenting with severe immunosuppression and hence denied the well-known benefits of life-saving ART in restoring their immune function.[21,22] Thus, the finding highlights the importance of ART in limiting the unfavourable treatment outcomes among MDR/RR-TB patients in high HIV burden countries.

A quarter of patients in our study encountered SAEs during their treatment and similar findings were reported in a recent meta-analysis among MDR/RR-TB patients in high HIV burden countries. However, the percentages varied from 13% to 43% in various studies included in the review. The settings where individualised drug regimens were used had reported lower percentages of SAEs. Relatively, the high percentage of SAEs in our study setting may be due to the fact that all the patients had received standardised SLDs.

There are some important implications arising from this study. First, despite MDR/RR-TB being less prevalent among new TB cases in comparison to those with recurrent TB, the proportion of new TB cases among the absolute number diagnosed with MDR/RR-TB in our study was much lower than the expected >90%. This coincides with the study period when the use of Xpert MTB/Rif testing was restricted only to high-risk populations and probably led to underdiagnoses of MDR/RR-TB among those with new TB. This has been addressed since the country adopted the use of Gene Xpert as the diagnostic test of choice among all presumptive TB patients in 2017.

Second, the high proportion of patients who did not have CDST results during their treatment is cause for concern as this is essential in monitoring bacteriological response to treatment. A recent study from Zimbabwe showed leakages in receipt of sputum samples at NRLs, culture contamination among received sputum specimens leading to a reduced proportion of samples with CDST results.[25] This CDST system will require improvements by including feedback of CDST results to facilities in order to inform patient management.

Third, we observed higher treatment success rates in comparison to what is reported to the WHO by the NTP of Zimbabwe.[1] The low treatment success rate reported by the NTP is a result of the high proportion of 'not evaluated' (32%) among unfavourable outcomes. The 'not evaluated' was less than 10% in the current study, where the patient records were systematically assessed. This highlights the deficiency in the routine reporting process and the NTP needs to ensure that district TB co-ordinators evaluate and report all treatment outcomes as recorded on patient 'treatment cards'.

Fourth, the high occurrence of SAEs in our study is also documented in other literature on the use of standardized MDR-TB regimens.[5,17] The NTP needs to institute standard protocols for monitoring, recording, reporting and management of SAEs. Though the PMDT guidelines provide direction for the management of SAEs, the adherence to these guidelines has not been systematically evaluated. Currently, there is no standardized documentation of SAEs on

the patient 'treatment cards'. Thus, the treatment cards need to be revisited to include space for structured documentation of SAEs and their management. The programme officers during their supervisory visits can review the 'treatment cards' for monitoring the management of SAEs. Also, the NTP could adopt individualized shorter MDR-TB treatment regimens, which has been shown to have fewer SAEs and treatment outcomes similar to standardized SLDs. [26–28]

Fifth, while there was a high uptake of ART among those HIV co-infected, there is a need to ensure all MDR/RR-TB patients diagnosed with HIV are initiated timely on ART in order to lessen the risk of having unfavourable outcomes. This attention should be particularly targeted at those newly diagnosed with MDR/RR-TB who are less likely to know their HIV status upon presentation with presumptive TB.

Last, a significant number of patients delayed initiation of MDR-TB treatment by more than 30 days after RR-TB diagnosis. While this was not associated with having an unfavourable outcome, this has got direct consequences at an individual level towards disease progression and at a population-level towards MDR-TB transmission in the community.

In conclusion, our findings show suboptimal MDR/RR-TB treatment success rates in this largely HIV co-infected patient population. Ensuring ART uptake among those ART-naïve patients could help in improving treatment success rates. Adoption of the new shorter treatment regimen should be considered, in view of the high occurrence of SAEs in this study. Future studies should focus on profiling management of MDR/RR-TB patients accessing care at the primary healthcare facilities in this setting.

## Acknowledgments

This research was conducted through the Structured Operational Research and Training Initiative (SORT IT), a global partnership led by the Special Programme for Research and Training in Tropical Diseases at the World Health Organization (WHO/TDR). The training model is based on a course developed jointly by the International Union Against Tuberculosis and Lung Disease (The Union) and Medécins sans Frontières (MSF). The specific SORT IT program which resulted in this publication was implemented by the Centre for Operational Research, The Union, Paris, France. Mentorship and the coordination/facilitation of this particular SORT IT workshop was provided through the Centre for Operational Research, The Union, Paris, France; the Department of Tuberculosis and HIV, The Union, Paris, France; the University of Washington, School of Public Health, Department of Global Health, Seattle, Washington, USA; National Institute for Medical Research, Muhimbili Centre, Dar es Salaam, Tanzania; and AIDS & TB Department, Ministry of Health & Child Care, Harare, Zimbabwe

The contribution by the District TB coordinators who collected the data and completed the proforma is well appreciated.

## Author Contributions

**Conceptualization:** Ronnie Matambo, Charles Sandy, Elliot Chikaka, Shungu Mutero-Munyati.

**Formal analysis:** Ronnie Matambo, Kudakwashe C. Takarinda, Pruthu Thekkur, Sungano Mharakurwa, Talent Makoni, Ronald Ncube, Kelvin Charambira, Christopher Zishiri, Mkhokheli Ngwenya, Albert Chiteka.

**Investigation:** Ronnie Matambo, Sungano Mharakurwa, Ronald Ncube, Saziso Nyathi, Albert Chiteka, Elliot Chikaka, Shungu Mutero-Munyati.

**Methodology:** Ronnie Matambo, Kudakwashe C. Takarinda, Pruthu Thekkur, Charles Sandy, Talent Makoni, Kelvin Charambira, Christopher Zishiri, Mkhokheli Ngwenya, Saziso Nyathi.

**Project administration:** Ronnie Matambo, Charles Sandy, Albert Chiteka, Shungu Mutero-Munyati.

**Supervision:** Ronnie Matambo, Charles Sandy, Sungano Mharakurwa, Christopher Zishiri, Shungu Mutero-Munyati.

**Validation:** Ronnie Matambo, Pruthu Thekkur, Sungano Mharakurwa, Talent Makoni, Mkhokheli Ngwenya.

**Writing – original draft:** Ronnie Matambo, Charles Sandy, Talent Makoni, Ronald Ncube, Kelvin Charambira, Mkhokheli Ngwenya, Saziso Nyathi, Albert Chiteka, Shungu Mutero-Munyati.

**Writing – review & editing:** Ronnie Matambo, Kudakwashe C. Takarinda, Pruthu Thekkur, Sungano Mharakurwa, Elliot Chikaka.

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
