## [Decision Letter · Decision Letter 0]

20 Aug 2019

PONE-D-19-16150

Treatment outcomes of Multi drug resistant and Rifampicin resistant Tuberculosis in Zimbabwe: A cohort analysis of patients initiated on treatment during 2010 to 2015

PLOS ONE

Dear Mr Matambo,

Thank you for submitting your manuscript to PLOS ONE. After careful consideration, we feel that it has merit but does not fully meet PLOS ONE’s publication criteria as it currently stands. Therefore, we invite you to submit a revised version of the manuscript that addresses the points raised during the review process.

We would appreciate receiving your revised manuscript by Oct 04 2019 11:59PM. To enhance the reproducibility of your results, we recommend that if applicable you deposit your laboratory protocols in protocols.io, where a protocol can be assigned its own identifier (DOI) such that it can be cited independently in the future. For instructions see: http://journals.plos.org/plosone/s/submission-guidelines#loc-laboratory-protocols

We look forward to receiving your revised manuscript.

Kind regards,

Denise Evans, PhD

Academic Editor

PLOS ONE

Journal Requirements:

2. In ethics statement in the manuscript and in the online submission form, please provide additional information about the patient records/samples used in your retrospective study. Specifically, please ensure that you have discussed whether all data/samples were fully anonymized before you accessed them and/or whether the IRB or ethics committee waived the requirement for informed consent. If patients provided informed written consent to have data/samples from their medical records used in research, please include this information.

3. Please note that all PLOS journals ask authors to adhere to our policies for sharing of data and materials: https://journals.plos.org/plosone/s/data-availability. According to PLOS ONE’s Data Availability policy, we require that the minimal dataset underlying results reported in the submission must be made immediately and freely available at the time of publication. As such, please remove any instances of 'unpublished data' or 'data not shown' in your manuscript and replace these with either the relevant data (in the form of additional figures, tables or descriptive text, as appropriate), a citation to where the data can be found, or remove altogether any statements supported by data not presented in the manuscript

4. Our internal editors have looked over your manuscript and determined that it is within the scope of our Antimicrobial Resistance call for papers. This collection of papers is headed by a team of Guest Editors for PLOS ONE: Kathryn Holt (Monash University and London School of Hygiene and Tropical Medicine), Alison H. Holmes (Imperial College London), Alessandro Cassini (WHO Infection Prevention and Control Global Unit), Jaap A. Wagenaar (Utrecht University). The Collection will encompass a diverse range of research articles; additional information can be found on our announcement page: https://collections.plos.org/s/antimicrobial-resistance. If you would like your manuscript to be considered for this collection, please let us know in your cover letter and we will ensure that your paper is treated as if you were responding to this call. If you would prefer to remove your manuscript from collection consideration, please specify this in the cover letter.

6. We note that you have indicated that data from this study are available upon request. PLOS only allows data to be available upon request if there are legal or ethical restrictions on sharing data publicly. For information on unacceptable data access restrictions, please see http://journals.plos.org/plosone/s/data-availability#loc-unacceptable-data-access-restrictions.

7. We note that you have included the phrase “data not shown” in your manuscript. Unfortunately, this does not meet our data sharing requirements. PLOS does not permit references to inaccessible data. We require that authors provide all relevant data within the paper, Supporting Information files, or in an acceptable, public repository. Please add a citation to support this phrase or upload the data that corresponds with these findings to a stable repository (such as Figshare or Dryad) and provide and URLs, DOIs, or accession numbers that may be used to access these data. Or, if the data are not a core part of the research being presented in your study, we ask that you remove the phrase that refers to these data.

Additional Editor Comments:

This paper describes the treatment outcomes of drug-resistant TB in Zimbabwe.

In addition to the Reviewer comments, I have the following comments that the authors should consider.

1. The authors should consider revising their concluding sentence which reads "There is a need for increased uptake of ART". HIV status and ART status was reported at initiation of treatment. Some of the patients who are HIV positive may be newly diagnosed and may not have initiated ART yet - it would be useful to look and see if those who were HIV positive but not on ART at treatment initiation, initiated ART between 2 weeks to 2 months depending on the CD4 count (as per guidelines). You may find some of the HIV positive patients initiated ART after they initiated DR-TB, and therefore it is difficult to make inferences about treatment outcomes.

2. Line 120 - I find the use of "short-course" confusing here, since the WHO has guidelines for short- and long-course DR-TB treatment. I would consider rephrasing. The reference [2] "DOTS-Plus and Green Light Committee" refers to 6-8 months with first-line anti-TB treatment.

3. Line 134 - requires a reference for "WHO has recommended shorter regimens".

4. Line 150 - Please clarify what the short course regimen is, in your setting (e.g. shorter, injectable-based, 9-12 month regimen)

4. Line 132 - The authors refer to the shorter regimen and Line 152 - authors state that the short regimen has been rolled out in the NTP. Since the paper focuses on the standard long-course RR/MDR-TB regimen (18-24 month), between 2010 - 2015, the authors need to mention why the paper is relevant (since short course has been adopted by the NTP).

5. Line 189 - write out "MTB/Rif" the first time it appears in the text; Line 192 - write out "PTB" the first time it appears in the text e.g. pulmonary TB (PTB); Line 224 - write out "PMDT" the first time it appears in the text; Line 230-233 - write out plus (+) and Lopinavir/Ritonavir and the correct abbreviation LPV/r

6. The authors have introduced selection bias, which is a limitation of the study. The study includes MDR/RR-TB patients who initiated treatment between 2010 and 2015 and continued treatment at either district hospitals or urban polyclinics (Line 237). Those who were referred to primary health facilities were excluded (Line 239). This may contribute to why treatment outcomes are different to what is reported to the WHO by the NTP of Zimbabwe. Those who are referred to primary health facilities are considered more stable, tolerating the regimen, and live close to the PHC. These factors, including distance, have been shown to be associated with treatment outcomes, adherence to treatment and utilization of treatment services. The study population therefore presents patients who were unstable/not tolerating the regimen and/or those who lived >10km from the health facility. The fact that the study does not include any treatment outcomes for patients referred to primary health facilities is a limitation.

7. Line 219 - please clarify what constitutes "stable". What criteria is used to determine if a patient is stable?

8. Line 252 - Please clarify how "missed doses" is reported. Is this patient self-report or obtained from clinical data (e.g. dispensed)?

9. Line 269 - regarding treatment outcomes; (1) authors should refer to the WHO reporting framework and include the definitions of the outcomes, (2) authors needs to include when the outcome was defined (i.e. how long were patients followed-up for, what is the person-time?), (3) the definition of primary outcome seems unusual - can authors include a reference for this?, (4) since death and LTFU are assigned when the outcome occurs, it is more appropriate to combine death + LTFU in your outcome definition and use Cox Proportional Hazard regression

10. Line 280 - 282 - IRB protocol numbers need to be included. Line 283 - "health and child care" should be corrected as in Line 282 "Ministry of Health and Child Care"

11. Line 233 - Guidelines for the use of cotrimoxazole (CPT) should be included in the Methods. Please explain what this variable represents. Is this on CPT at treatment initiation or a history of CPT?

12. Line 569 - some key variables are missing in Table 2. Examples include resistance pattern (e.g. MDR-TB, RR-TB (mono; isoniazid sensitive) or RR-TB with additional resistance unknown), time on ART, CD4 count, smear microscopy results, weight/BMI (Line 268), EPTB vs. PTB.

13. Line 236 and Table 1 - please clarify if the study includes children? From Table 1 n=24 were <24 years. It is not clear if this is 18-24 or 0-24? If the latter, then this should be further categorized as children (<10) or adolescents (10-24) or further as young adolescent (10-14), older adolescent (15-19) and young adult (20-24).

14. Table 2 Encountered SAEs - Is this during treatment or at treatment initiation. If the former, then this should be separated out from the characteristics at treatment initiation (Clarify that Table 2 includes clinical characteristics AT treatment initiation). Would be useful to include/mention the most common SAEs reported in the Table (i.e. Table 3?)

15. Table 3 Outcomes should add up to 100% - Failed should be 0.9% and not <1 and LTFU 8.2% (39/473). Therefore, 26.4 + 8.2 + 0.9 + 3.4 = 100%

16. Table 4 - consider defining a resistance pattern variable - which should encompass the diagnostic test and the DST results.

17. Table 4 - The current model ignores the time-varying nature of adherence.

18. Figure 1 - It would be useful to present outcomes as typically reported by the WHO (e.g. see https://www.who.int/tb/publications/global_report/gtbr2017_main_text.pdf - Page 80)

Reviewers' comments:

Reviewer's Responses to Questions

**Comments to the Author**

1. Is the manuscript technically sound, and do the data support the conclusions?

Reviewer #1: Yes

Reviewer #2: Yes

Reviewer #3: No

2. Has the statistical analysis been performed appropriately and rigorously? 

Reviewer #1: I Don't Know

Reviewer #2: Yes

Reviewer #3: No

3. Have the authors made all data underlying the findings in their manuscript fully available?

Reviewer #1: Yes

Reviewer #2: Yes

Reviewer #3: Yes

4. Is the manuscript presented in an intelligible fashion and written in standard English?

Reviewer #1: Yes

Reviewer #2: Yes

Reviewer #3: Yes

5. Review Comments to the Author

Reviewer #1: I appreciate the work of the Authors to try to define factors associated with unfavourable outcomes in RR and MDR TB in a Country with a very high percentage of HIV co-infected patients.

I suggest the Authors to stress the need for individualized treatment regimens in order to decrease side effects (BMC Infect Dis. 2019 Jun 28;19(1):564. doi: 10.1186/s12879-019-4211-0.) and to cite the need for better treatment compliance (BMC Infect Dis. 2017 Jan 21;17(1):91. doi: 10.1186/s12879-017-2200-8. and Multidiscip Respir Med. 2018 Nov 9;13:41. doi: 10.1186/s40248-018-0154-3. eCollection 2018)

Reviewer #2: TITLE – Consider adding ‘Urban Setting’ to the title since the population belongs to urban areas and probably a bias to favorable outcome.

Abstract – appropriate – consider elaborating the methods.

Introduction – Describes the problem in a complete and relevant manner. Aim of the study is well reported.

Method –

Please consider mentioning the type of phenotypic cultures utilized.

Operational definition of treatment outcomes, although standardized as per WHO – please reiterate here, this can prevent a bit of confusion

Results –

Now as mentioned by the authors in terms of limitation – the absence of data on management of SAE, CDST results, CD4 count and comorbidity data. I can suggest certain key data variables which are missing, if available, consider providing them.

BMI, radiological evidence of cavities as they may have direct effect on outcome as well as occupational history and smoking status of the cohort, both of which are strong risk factors for developing TB and unfavorable outcome.

It is unfortunate that DST results of fluoroquinolones and second line injectables are not available, both of which are known to be risk factors for adverse outcome, if for the ~53 cases this data can be made available and analysed as a risk factor, it will greatly strengthen your study. It is a strong point for advocating the need of DST.

Data for Time to unfavourable outcome – is a strong variable that can help us in understanding the severity of the cohort population and help in future intervention. Did any SAE led to unfavourable outcome? The individuals who died or lost to follow up – were recorded to be failing the regimen?

Details of ART initiation is not provided neither the status of ART success or failing is analysed.

Discussion:

The authors have done proper analysis of their results, aptly described the limitation and strengths of the study as well as compared the results with similar studies.

Please elaborate on the reason for 180 people with no recorded time for culture conversion – they failed to culture convert or some of them died before or some could not give the sample. How many patient had their sputum culture status reverted back to positive?

Interestingly, ~14% of the study cohort was initiated on ATT beyond 30 days of RRTB diagnosis, was not found to affect the outcome. You may consider highlighting that delay in initiation for SLD.

Reviewer #3: Please see my comments which have been attached as Microsoft Word Document. I have included all my comments in this document.

6. PLOS authors have the option to publish the peer review history of their article (what does this mean?). If published, this will include your full peer review and any attached files.

Reviewer #1: No

Reviewer #2: No

Reviewer #3: No

---

## [Author Response · Author response to Decision Letter 0]

14 Oct 2019

28 August 2019

PLOS ONE

Dear Editor in Chief

Please find below a reply letter to our submission, titled:

PONE-D-19-16150: Treatment outcomes of Multi drug resistant and Rifampicin resistant Tuberculosis in Zimbabwe: A cohort analysis of patients initiated on treatment during 2010 to 2015

We thank the reviewers and the Editor for their valuable comments to our initial submission. We have attempted to address all the queries made. We provide below a point-by-point response to the queries raised and we have also provided both the clean version of the manuscript and a document with track changes.

Editor comments

Comment #1

Response

Noted thank you. This has been addressed as per requirement

Comment #2

In ethics statement in the manuscript and in the online submission form, please provide additional information about the patient records/samples used in your retrospective study. Specifically, please ensure that you have discussed whether all data/samples were fully anonymized before you accessed them and/or whether the IRB or ethics committee waived the requirement for informed consent. If patients provided informed written consent to have data/samples from their medical records used in research, please include this information.

Response

Thank you for raising this very important ethical issue. Data was fully anonymised as only unique identifiers in the form of TB registration numbers were abstracted onto the data collection proforma in place of patient names. The Ethics committee waived the requirement for individual informed consent since this was a retrospective study and the Ministry of Health and Child Care had given permission to access patients clinical records at participating centres.

Comment #3

Please note that all PLOS journals ask authors to adhere to our policies for sharing of data and materials: https://journals.plos.org/plosone/s/data-availability. According to PLOS ONE’s Data Availability policy, we require that the minimal dataset underlying results reported in the submission must be made immediately and freely available at the time of publication. As such, please remove any instances of 'unpublished data' or 'data not shown' in your manuscript and replace these with either the relevant data (in the form of additional figures, tables or descriptive text, as appropriate), a citation to where the data can be found, or remove altogether any statements supported by data not presented in the manuscript

Response

Thank you for raising this. The data not shown on distribution and type of severe adverse events has now been added as Table 3.

Comment #4

 Our internal editors have looked over your manuscript and determined that it is within the scope of our Antimicrobial Resistance call for papers. This collection of papers is headed by a team of Guest Editors for PLOS ONE: Kathryn Holt (Monash University and London School of Hygiene and Tropical Medicine), Alison H. Holmes (Imperial College London), Alessandro Cassini (WHO Infection Prevention and Control Global Unit), Jaap A. Wagenaar (Utrecht University). The Collection will encompass a diverse range of research articles; additional information can be found on our announcement page: https://collections.plos.org/s/antimicrobial-resistance. If you would like your manuscript to be considered for this collection, please let us know in your cover letter and we will ensure that your paper is treated as if you were responding to this call. If you would prefer to remove your manuscript from collection consideration, please specify this in the cover letter.

Response

Thank you for considering our manuscript for the PLOS ONE collection on Antimicrobial Resistance. We would like to express our interest to have our manuscript considered as such and we give you explicit permission to process accordingly. We have indicated our interest in the attached cover letter

Comment #6

 We note that you have indicated that data from this study are available upon request. PLOS only allows data to be available upon request if there are legal or ethical restrictions on sharing data publicly. For information on unacceptable data access restrictions, please see http://journals.plos.org/plosone/s/data-availability#loc-unacceptable-data-access-restrictions.

Response

Thank you for the comment. Unfortunately we are not permitted to share our dataset due to our local ethics clearance committee and the Ministry of Health and Child Care who are the custodians of all data related to the National TB Programme. We have provided details of institutional heads for both organizations in the cover letter as requested.

Comment #7

We note that you have included the phrase “data not shown” in your manuscript. Unfortunately, this does not meet our data sharing requirements. PLOS does not permit references to inaccessible data. We require that authors provide all relevant data within the paper, Supporting Information files, or in an acceptable, public repository. Please add a citation to support this phrase or upload the data that corresponds with these findings to a stable repository (such as Figshare or Dryad) and provide and URLs, DOIs, or accession numbers that may be used to access these data. Or, if the data are not a core part of the research being presented in your study, we ask that you remove the phrase that refers to these data.

Response

The data that were not shown have now been added as “Table 3”.

Additional Editor Comments:

Comment #1

In addition to the Reviewer comments, I have the following comments that the authors should consider.

1. The authors should consider revising their concluding sentence which reads "There is a need for increased uptake of ART". HIV status and ART status was reported at initiation of treatment. Some of the patients who are HIV positive may be newly diagnosed and may not have initiated ART yet - it would be useful to look and see if those who were HIV positive but not on ART at treatment initiation, initiated ART between 2 weeks to 2 months depending on the CD4 count (as per guidelines). You may find some of the HIV positive patients initiated ART after they initiated DR-TB, and therefore it is difficult to make inferences about treatment outcomes.

Response

Thank you. In our study, dates on when ART was initiated in relation to commencement of TB treatment were not collected. However, the ART status of those who newly tested HIV-positive are supposed to be updated in the DR-TB register throughout their course of MDR-TB treatment period together with date of ART initiation although the information of ART start dates were missing in our case. As a result this limits our capabilities to determine impact of ART timing on DR-TB treatment outcomes, however we feel there is merit to conclude that ART does improve DR-TB treatment outcomes. Instead we have added to the abstract the following sentence: “Assessing timing of ART initiation in relation to TB treatment outcomes will also require future exploration.” We hope this response is satisfactory.

Comment #2

2. Line 120 - I find the use of "short-course" confusing here, since the WHO has guidelines for short- and long-course DR-TB treatment. I would consider rephrasing. The reference [2] "DOTS-Plus and Green Light Committee" refers to 6-8 months with first-line anti-TB treatment.

Response

Noted, the term “Short course” has been removed and the sentence now reads as follows: “In 2000, WHO recommended the 20-24 month standardized second-line drug (SLDs) regimens for treatment of MDR/RR-TB patients in resource-limited settings”. We have also removed the reference [2] on the DOTS-Plus and the Green Light Committee.

Comment #3

3. Line 134 - requires a reference for "WHO has recommended shorter regimens".

Response

Thank you. We have now referenced the 2019 WHO consolidated guidelines on drug-resistant tuberculosis treatment.

Comment #4

4. Line 150 - Please clarify what the short course regimen is, in your setting (e.g. shorter, injectable-based, 9-12 month regimen)

Response

Noted, thank you. Now clarified and it reads: “In 2016, the short treatment regimen which is over 9-11 months and injectable based was adopted”

Comment #4 (Duplication of numbering)

 Line 132 - The authors refer to the shorter regimen and Line 152 - authors state that the short regimen has been rolled out in the NTP. Since the paper focuses on the standard long-course RR/MDR-TB regimen (18-24 month), between 2010 - 2015, the authors need to mention why the paper is relevant (since short course has been adopted by the NTP).

Response

The short treatment course was piloted since June 2018 and is made available only in a few districts. Even in districts where short regimen is available, very few patients are started on it due to issues in assessing eligibility. Up to now most districts are still implementing the 18-24 month regimen. The short regimen might have been initiated in less than 5% patients. The study is relevant as it informs on the treatment outcomes with longer regimens and also provides a baseline estimate for tracking the successful outcomes rates with scale up of short regimen in future. 

Comment #5

5. Line 189 - write out "MTB/Rif" the first time it appears in the text; Line 192 - write out "PTB" the first time it appears in the text e.g. pulmonary TB (PTB); Line 224 - write out "PMDT" the first time it appears in the text; Line 230-233 - write out plus (+) and Lopinavir/Ritonavir and the correct abbreviation LPV/r

Response

Line 189-"MTB/Rif" now appears in full the first time it appears in the text

Line 192-“PTB” now appears as “Pulmonary TB” the first time it appears in the text

Line 224-“PMDT" written in full the first time it appeared in line 149.

Line 230-233- Lopinavir/Ritonavir) (ABC + 3TC + EFV now reflected and LPV/r now reflected

Comment #6

6. The authors have introduced selection bias, which is a limitation of the study. The study includes MDR/RR-TB patients who initiated treatment between 2010 and 2015 and continued treatment at either district hospitals or urban polyclinics (Line 237). Those who were referred to primary health facilities were excluded (Line 239). This may contribute to why treatment outcomes are different to what is reported to the WHO by the NTP of Zimbabwe. Those who are referred to primary health facilities are considered more stable, tolerating the regimen, and live close to the PHC. These factors, including distance, have been shown to be associated with treatment outcomes, adherence to treatment and utilization of treatment services. The study population therefore presents patients who were unstable/not tolerating the regimen and/or those who lived >10km from the health facility. The fact that the study does not include any treatment outcomes for patients referred to primary health facilities is a limitation.

Response

Thank you very much for concurring with the authors that the study had some limitations as referred above. The above Editor concern has been articulated at length as a study limitation and authors acknowledged this quite extensively

Comment #7

7. Line 219 - please clarify what constitutes "stable". What criteria is used to determine if a patient is stable?

Response

“Stable” is a term used to describe those patients who were able to ingest medication, did not show signs of adverse drug reaction and had all the laboratory investigations within normal limit. This description now appears in line 219

Comment #8

8. Line 252 - Please clarify how "missed doses" is reported. Is this patient self-report or obtained from clinical data (e.g. dispensed)?

Response

“Missed doses” are determined from patient clinical records specifically from the patient treatment card. As the directly observed treatment is provided, the ‘right tick’ is made against the particular date in the treatment card by the clinician when drugs are consumed. In case if the patient fails to report for DOT at the health facility, a follow up is done and in the event the patient is not located and misses his drugs for that day, it is marked as missed with ‘wrong’ tick against that date. We considered all the ‘wrong’ tick till the date of treatment outcome as ‘Missed doses’, Later we calculated the percentage of missed doses per total number of days the patient was receiving the treatment.

Comment #9

9(1) Line 269 - regarding treatment outcomes; (1) authors should refer to the WHO reporting framework and include the definitions of the outcomes.

Response

Noted. “Table of Treatment Outcomes definition” referred as Fig 2

9(2) authors needs to include when the outcome was defined (i.e. how long were patients followed-up for, what is the person-time?), (3) the definition of primary outcome seems unusual - can authors include a reference for this?, (4) since death and LTFU are assigned when the outcome occurs, it is more appropriate to combine death + LTFU in your outcome definition and use Cox Proportional Hazard regression

Response

Patient TB treatment outcomes were defined at 24 months from date of each patients MDR-TB treatment start date until date of their respective outcomes for deaths, LTFUs, treatment failure and “outcome not evaluated”. Those classified as LTFU were defined as a patient whose MDR-TB treatment was interrupted for two or more consecutive months for any reason as per WHO guidelines. Those cured or who completed their treatment were categorized as successful treatment outcomes, whereas the others were categorized as unsuccessful treatment outcomes in line with recent meta-analysis studies by Kibret et al and Johston JC et al. We have added this to the data analysis section of our paper

The 473 patients in our study contributed 672.5 person-years. We have added this to the first paragraph of the results section. Thank you for suggesting that we use Cox Proportional Hazards regression. We considered this at length as the most appropriate analysis method, however we finally abandoned it because there were a number of patients who had missing dates on treatment outcomes hence this would lower our sample size.

Comment #10

10. Line 280 - 282 - IRB protocol numbers need to be included. Line 283 - "health and child care" should be corrected as in Line 282 "Ministry of Health and Child Care"

Response

Line 280 - 282 - IRB protocol numbers now included.

Line 283 - "health and child care"- Both sentences line 282 and 283 read as “Ministry of Health and Child Care” from original texts

Comment #11

Line 233 - Guidelines for the use of cotrimoxazole (CPT) should be included in the Methods. 

Response

A reference for the guidelines of use of Cotrimoxazole Preventive Therapy is now indicated and the reference section has been updated

Comment #12

12. Line 569 - some key variables are missing in Table 2. Examples include resistance pattern (e.g. MDR-TB, RR-TB (mono; isoniazid sensitive) or RR-TB with additional resistance unknown), time on ART, CD4 count, smear microscopy results, weight/BMI (Line 268), EPTB vs. PTB.

Response

Thank you for observing that some key variables are missing in Table 2. All the above are missing data as explained in the section on study limitations

Comment #13

13. Line 236 and Table 1 - please clarify if the study includes children? From Table 1 n=24 were <24 years. It is not clear if this is 18-24 or 0-24? If the latter, then this should be further categorized as children (<10) or adolescents (10-24) or further as young adolescent (10-14), older adolescent (15-19) and young adult (20-24).

Response

The study does include children. We had collapsed the age groups <24 years because of small numbers, however we have reverted back to the disaggregation of <5, 5-14 and 15-24 years in Table 1 but however maintained the combined age group of <24 years in the multivariate regression analysis table so as to allow for adequate numbers for calculating relative risks and their 95% confidence intervals. We also used the age categories <24, 25-34, 35-44, 45-54 and 55+ years as these are the standard age groups for reporting TB data to WHO and are also used in several studies to allow for comparisons.

Comment #14

14. Table 2 Encountered SAEs - Is this during treatment or at treatment initiation. If the former, then this should be separated out from the characteristics at treatment initiation (Clarify that Table 2 includes clinical characteristics AT treatment initiation). Would be useful to include/mention the most common SAEs reported in the Table (i.e. Table 3?)

Response

Thank you for the point. The encountered SAEs were during treatment and we have rephrased the variable as “Encountered SAE during treatment”. Seeing that some patients may have also initiated ART during MDR-TB treatment and that we have the variable on missed treatment dosses in this table, we have instead rephrased the table title as “Clinical characteristics of MDR/RR-TB patients at baseline and/or during treatment among MDR/RR-TB patients initiated on treatment in Zimbabwe, 2010-2015”. We hope that this is satisfactory.

Comment #15

15. Table 3 Outcomes should add up to 100% - Failed should be 0.9% and not <1 and LTFU 8.2% (39/473). Therefore, 26.4 + 8.2 + 0.9 + 3.4 = 100%

Response

Many thanks for pointing this out. We have replaced the percentage for those with treatment failure with 0.8% however we note that the percentages for the different end-of-TB treatment outcomes add up to 99.9% due to rounding-off errors. We have specified this as a footnote to the table.

Comment #16

16. Table 4 - consider defining a resistance pattern variable - which should encompass the diagnostic test and the DST results.

Response

Thank you for the comment. This information was not available due to missing data on CDST results which the authors highlighted as a study limitation in the discussion section.

Comment #17

17. Table 4 – The current model ignores the time-varying nature of adherence.

Response

This is indeed a valid comment and we ought to have included missed doses as a time dependent variable in our univariate and multivariate-adjusted regression models. However, we did not have data on the specific time points when MDR-TB treatment doses were missed as these were aggregated for each patient by the time they achieved their treatment outcome

Comment #18

18. Figure 1 - It would be useful to present outcomes as typically reported by the WHO (e.g. see https://www.who.int/tb/publications/global_report/gtbr2017_main_text.pdf - Page 80)

Response

Thank you for the comment. The term “Unfavourable Outcome” has been further clarified in the text as per WHO reporting system (Death, Treatment failure, LTFU and Not Evaluated)

 Reviewer #1: 

I appreciate the work of the Authors to try to define factors associated with unfavourable outcomes in RR and MDR TB in a Country with a very high percentage of HIV co-infected patients.

I suggest the Authors to stress the need for individualized treatment regimens in order to decrease side effects (BMC Infect Dis. 2019 Jun 28;19(1):564. doi: 10.1186/s12879-019-4211-0.) and to cite the need for better treatment compliance (BMC Infect Dis. 2017 Jan 21;17(1):91. doi: 10.1186/s12879-017-2200-8. and Multidiscip Respir Med. 2018 Nov 9;13:41. doi: 10.1186/s40248-018-0154-3. eCollection 2018)

Response

The authors appreciate this valid comment. On study implications, line 409, the authors stresses the need for NTP to adopt shorter treatment regimen for MDR-TB treatment which has been shown to have fewer SAEs. A reference has been added and cited in the reference section as advised

Line 417, the authors stresses the need to ensure that patients do not have missed doses during MDR-TB treatment in order to lessen their risk of having unfavourable outcomes. . A reference has been added and cited in the reference section as advised.

 Reviewer #2: 

TITLE – Consider adding ‘Urban Setting’ to the title since the population belongs to urban areas and probably a bias to favorable outcome.

Response

Thank you for raising this important point, however the study was not primarily set for the urban population as some district hospitals serve mostly rural populations thus the study population was both urban and rural

Abstract – appropriate – consider elaborating the methods.

Response

Thank you for this comment. The methods section of the abstract has been elaborated as advised. The following statement has been added “A generalized linear model with a log-link and binomial distribution or a poisson distribution with robust error variances were used assess factors associated with “unfavorable outcome”. The unadjusted and adjusted relative risks were calculated as measure of association. A �value< 0.05 was considered statistically significant”.

Introduction – Describes the problem in a complete and relevant manner. Aim of the study is well reported.

Response

Thank you, well appreciated

Comment

Method – Please consider mentioning the type of phenotypic cultures utilized.

Response

Thank you. The type of phenotypic cultures utilized is now indicated as BBLTM MGITTM Mycobacterial Growth Indicator Tubes (Becton Dickinson, Sparks, MD

Comment

Operational definition of treatment outcomes, although standardized as per WHO – please reiterate here, this can prevent a bit of confusion

Response

 Noted, Fig 2 has been added to clarify

Comment

Results – Now as mentioned by the authors in terms of limitation – the absence of data on management of SAE, CDST results, CD4 count and comorbidity data. I can suggest certain key data variables which are missing, if available, consider providing them.

BMI, radiological evidence of cavities as they may have direct effect on outcome as well as occupational history and smoking status of the cohort, both of which are strong risk factors for developing TB and unfavorable outcome

Response

We thank you for raising this very valid comment. As already mentioned as study limitations, the data suggested was missing and cannot be included

Comment

It is unfortunate that DST results of fluoroquinolones and second line injectables are not available, both of which are known to be risk factors for adverse outcome, if for the ~53 cases this data can be made available and analysed as a risk factor, it will greatly strengthen your study. It is a strong point for advocating the need of DST.

Response

Thank you for the comment. Of all the second-line injectables and flouroquinolones in use (i.e. kanamycin, Levofloxacin, Moxifloxacin and Cycloserine and Capriomycin alone as an alternative to those intolerant to kanamycin), only 3 patients were resistant to any one of the above hence we could not include this variable in the univariate and multivariate regression analysis because of the small numbers.

Comment

Data for Time to unfavourable outcome – is a strong variable that can help us in understanding the severity of the cohort population and help in future intervention. Did any SAE led to unfavourable outcome? The individuals who died or lost to follow up – were recorded to be failing the regimen?

Response

Thank you for raising this very important observation. In our univariate and multivariate regression encountering an SAE was not significant predictor of an unfavourable outcome as shown in Table 5. Also, due to deficiencies in our data and this being a retrospective study, we failed to assess the cause-effect relationship of SAEs and unfavourable outcome

Comment

Details of ART initiation is not provided neither the status of ART success or failing is analysed.

Response

Thank you. Unfortunately data on virological, immunological or clinical failure of patients initiated on ART were not available in the DR-TB registers hence required matching patient names to the ART registers given that there was no unique identifier linking the ART and MDR-TB programmes. However we do acknowledge the importance of this data and is something that was worth exploring and should be explored in future studies.

Comment

Discussion:

The authors have done proper analysis of their results, aptly described the limitation and strengths of the study as well as compared the results with similar studies.

Response

Thank you, comment well appreciated

Comment

Please elaborate on the reason for 180 people with no recorded time for culture conversion – they failed to culture convert or some of them died before or some could not give the sample

Response

Thank you for asking us to elaborate. These patients either did not have their samples examined because of contamination, spillages or did not have results (CDST) sent back to the health facility and the reasons are well explained under the section on study implications where the authors mentioned that the high proportion of patients who did not have CDST results during their treatment is cause for concern as this is essential in monitoring bacteriological response to treatment. A recent study from Zimbabwe showed leakages in receipt of sputum samples at NRLs, culture contamination among received sputum specimens leading to a reduced proportion of samples with CDST results and this was mentioned in the relevant section. The authors concluded by stressing that this CDST system will require improvements including feedback of CDST results to facilities in order to inform patient management.

Comment

How many patient had their sputum culture status reverted back to positive

Response

Thank you for this question. None had their sputum culture status reverted back to positive

Comment

Interestingly, ~14% of the study cohort was initiated on ATT beyond 30 days of RRTB diagnosis, was not found to affect the outcome. You may consider highlighting that delay in initiation for SLD.

Response

Thank you for pointing this out. We have mentioned the following in the discussion section: “Last, a significant number of patients delayed initiation of MDR-TB treatment by more than 30 days after RR-TB diagnosis. Whilst this was not associated with having an unfavourable outcome, this has got dire consequences at an individual level towards disease progression and at a population-level towards MDR-TB transmission in the community.”

 Reviewer #3:

Comment #1

Background is unclear and does not adequately contextualize the aims of the manuscript.

Response

Thank you for the comment. This study aimed at assessing the profile, treatment outcomes and factors associated with unfavourable treatment outcomes among patients initiated on MDR/RR-TB treatment. In view of this we started by highlighting the global burden of MDR/RR-TB and also mentioned the WHO MDR/RR-TB treatment targets as a benchmark to assess our poor performance in this regard. We further summarised some of the literature of risk factors associated with poor MDR-TB treatment outcomes before we highlighted the paucity of data on treatment outcome determinants in our setting. This is of interest to the Zimbabwe NTP in order to have tailored interventions aimed at improving our overall MDR-TB treatment success rate. We do appreciate your comment and would like your further guidance on specific sections that require rewriting in order to make the introduction more appealing.

Comment #2

Methods lack clarity on how outcomes were defined, time-points at which outcomes were assessed by, measurement of covariates like adverse events is not suitable, inferential analytic techniques are also questionable.

Response

Thank you for this comment. We have addressed your comment on outcome definitions and time-points for assessment of these outcomes are addressed in the data entry and analysis section in line with other reviewers comments. As for inferential analytical techniques, we would have preferred to use Cox Hazards regression analysis to generate hazard ratios, however due to a large proportion of missing data on time of outcomes, we resorted to generating relative risk instead. Also, as our outcome of interest in this manuscript is ‘unfavourable outcome’, which includes ‘death’, LTFU, failure and ‘not evaluated’ we considered binomial regression. If at all we were modelling for death or LTFU than the Cox Hazard regression would have been the only option.

Comment #3

Results are sparse and could be described in greater detail.

Response

Thank you. In our analysis, we generated tables that responded to each of our set specific objectives and were hesitant to delve into greater detail for fear of digressing from our set objectives. However, we would appreciate if you can specify particular detail which would be of interest to the readers of our paper and is also aligned with our study objectives.

Comment #4

It is not very clear how background, aims, methods, results tie into the discussion and conclusions. 

Response

Thank you for the comment. We are however surprised that this is the case as your comment is contradictory to the other 2 reviewers. In writing the paper we carefully attempted to logically align our discussion to the results presented which in turn were informed by the background, aims and methods of the study. We hope this is understandable with you

Comments:

1. Keywords 

a. Repeated (full-text and abbreviations used for the same word)

Response

We have gone through the whole manuscript and ensured that full-text and abbreviations are reported together only once, following which only abbreviations are mentioned were necessary.

a) Background 

Comment

Tuberculosis to be written out in full before abbreviated

Response

Thank you. Tuberculosis now written out in full the first time it is used before being abbreviated

Comment

Incorrect order of abbreviations for RR/MDR-TB. Authors write this out in full starting with MDR-TB.

Response

Thank you for this. We appreciate that MDR/RR-TB can be written as RR/MDR-TB however after careful consideration, the study investigator team opted to align with MDR/RR-TB which is used in the WHO global TB reports and guidelines. We hope this is understandable.

Comment

Please rephrase last sentence. This is unclear

Response

Thank you for the comment. The last sentence on the background now rephrased and reads “The profile, management, and factors associated with unfavourable treatment outcomes of MDR/RR TB have not been systematically evaluated in Zimbabwe”.

Comment

b) Design

1) Please provide more detail regarding the methods you employ to assess treatment outcomes and risk factors for such outcomes. Study sites, participants, data collected, analytic methods used, definition of outcomes assessed, time point outcomes were assessed by etc.

Response

Study sites were all district hospitals and urban polyclinics in Zimbabwe. The study participants were all MDR/RR-TB patients initiated on treatment between 2010-2015 under the Zimbabwe NTP and continued their treatment in the above mentioned health facilities. This information is stated in the section on “Study Population”. We have further on added the total number of district hospitals and urban polyclinics which totalled eighty-two under study design. As per your earlier request we have specified the time point outcomes assessed in the “Data entry and analysis” section.

2) How were participants selected for the final analysis?

Response

We thank you for asking us to explain. Of the total 935 MDR/RR-TB patients initiated on treated during study reference period , the 473 (51%) of patients who met the study inclusion criteria of those started on MDR/RR-TB treatment and continued their 20-24 month treatment at district and urban polyclinics were all included in the study

Comment

c) Results 

i. How was missing MDR-TB treatment doses defined? Please add to methods.

Response

Thank you. Percentage of missed doses is defined as percentage of the number of days with missed doses divided by the total number of days a patient was on treatment up until date of outcome. This has been added to methods section

Comments

ii. Did outcomes differ between RR-TB and MDR-TB?

Response

Thank you for this valid comment. As shown in Table 5, based on the individual analysis of isoniazid, streptomycin and ethambutol resistant patterns at baseline, there were no differences in outcomes between RR-TB and MDR-TB patients. Furthermore, it is difficult to conclusively distinguish between RR-TB and MDR-TB patients as the majority of patients did not have DST results recorded or at least for all second-line TB drugs..

Comment

iii. How were covariates selected to be included in the main effects model? Please add to methods.

 Response

 Thank you for the comment. All covariates with a p<0.25 and those that are biologically plausible predictors of unfavourable outcomes were added to the multivariate regression model. This is mentioned in the “Data entry and analysis” section.

Comment

iv. Why was ‘not evaluated’ outcome considered unfavourable?

Response 

Those “not evaluated” were included in those with unfavourable outcomes as stated in the “Data entry and analysis” section in line with recent meta-analysis studies by Kibret et al and Johston JC et al. The WHO in its annual TB report, considers ‘not evaluated’ as ‘unfavourable’ or ‘unsuccessful’ outcome

Comment

d)Conclusion

v. Stating that outcomes were poor coupled with high occurrence of adverse events make it seems like this was a significant risk factor – but it wasn’t?

Response 

Thank you for the comment. Whilst we acknowledge that adverse events were not a significant risk factor, in paragraph 6 of the discussion section we only highlight that the high proportion of SAEs in this patient cohort (i.e. one in four patients) is a cause for concern and do not insinuate that this is a risk factor for unfavourable outcomes. We further advocate for more standardized recording and reporting of SAEs for better tracking by the NTP and also advocate for wider adoption of the shorter MDR-TB treatment regimen

Comment

vi. Conclusions and recommendations are based on adverse events; however this is not mentioned as a significant predictor in the results. Therefore, how are these conclusions relevant to your findings?

Response 

Thank you. Please refer to the earlier response above.

2. Introduction -Comment

i. Capitalize ‘tuberculosis’ line 112

Response 

“T” Now in capital letters

Comment 

ii. Don’t start a sentence with an abbreviation – line 114 and other places within the text

Response 

 Line 114 and 118 corrected

Comment 

Insert “The” before World Health Organisation

Response

The World Health Organization’ – line 116

“The” Inserted

Comment

iii. Line 120 – is this reference from 2000?

Response

Thank you. Yes, we can confirm that the reference is from 2000

Comment

iv. Line 128-129 – please clarify what the second part of this sentence means

Response

The authors of the systematic review paper stated that studies included in the review were largely from countries with low HIV coinfection. They then recommended for systematic assessment of treatment outcomes in high HIV coinfection countries of sub-Saharan Africa

Comment

v. Could you comment on the incidence of ADRs on second-line TB treatment? Line 130-134

Response

Thank you for raising this comment. We could not generate incidence rate of ADRs as we did not have data on time of their occurrence.

Comment

vi. Could you comment on eligibility for newer shorter regimens? Line 130-134

Response

Thank you for asking us to comment. Eligibility for newer shorter regimens is when there is no risk or confirmed resistance to any second line medicines or any medicine used in the shorter treatment regimen

Comment

vii. However, unsure of commenting on newer short course regimens – what does this have to do with assessing treatment outcomes in your traditional-long course cohort? Shorter regimens may be more pertinent in your discussion.

Response

Our apologies as we do not clearly understand your comment. If we understand well, you suggest that we mention the importance of the newer short course regimen as an alternative to the traditional long course regimen. We have touched on this in paragraph 10 of the discussion section were we suggest that the NTP should adopt wider scale up of the shorter treatment regimen which has fewer SAEs although treatment outcomes are comparable to the long MDR-TB treatment regimen.

3. Methods

Comment 

viii. Could a figure describing the general setting be included to better understand the public healthcare setting? And thereafter selection process 

Response 

Thank you for suggesting this. We have attempted to present this information graphically but have concluded that the text description mentioned in the General setting and the description of diagnosis and treatment of MDR/RR-TB gives an overall description of Zimbabwe’s public health setting in the context of the National TB Programme. We are happy to clarify any specific sections that are unclear to you.

Comment

ix. Line 212-213 – WHO recommended DOTS-plus during the study period? Or as above in 2000?

Response

In 2000, Tense changed to reflect “Was” instead of “be”

Comment

x. Line 237 -238 please rephrase

Response

Rephrased as per reviewer 3 comments, sentence now short and more precise

Comment

xi. Would patients referred for DOTS-plus who were then excluded from the analysis be systematically different to those who were included?

Response

 Yes we do acknowledge this possibility given that the treatment success rate for our study population was better than that reported by the NTP for all patients due to their high proportion of unevaluated patients. Furthermore in rural settings health facilities are less accessible due to long distances from households and there is less specialized care in rural facilities. We have stated this as a limitation in the discussion section.

Comment

xii. Line 243 – consistency with capitalization

Response

We have noted this and addressed it accordingly.

Comment

xiii. Data quality between sites could differ? Were there any systematic differences between quality (missing data) etc. between sites?

Response 

Thank you for this comment, however there were no systematic differences between sites in terms of data quality. Missing data was across all sites because the tools used by all sites were the same and no data was collected on those variables. Also all the sites sent their samples to the same reference laboratory which did not give results back on CDST. Furthermore 10%of the data was subjected to quality assurance and originally missing data was completed before final data entry. 

Comment

xiv. Since the primary outcome of interest is death and LTFU which can occur any time from treatment initiation – would you consider using a Cox Hazard Model to determine hazard ratio estimates of ‘unfavourable outcomes’? Considering time to outcome may be an important factor which could be affected differently by different risk factors. Furthermore, was LTFU and death separated and did risk factors differ by these outcomes? Lastly, what is the motivation for including ‘not evaluated’ as an ‘unfavourable outcome’?

Response

 As mentioned above we could not perform Cox Hazard regression analysis due to the high proportion of missingness on time to event data. We also reserved further comparisons on any differences between predictors of mortality and LTFU for another paper as we felt this was too much information for one paper. We hope reviewers understand our position.

Comment

xv. Please include a time-point at which outcomes were assessed by. Did everyone have a minimum follow-up etc.?

Response

 Yes, since this was a retrospective cohort study based on routinely collected programme data, all patients were allowed a follow-up period for their potential 20-24 month MDR-TB treatment duration.

Comment

3Results 

xvi. Line 287 – punctuation

Response 

Thank you, we consider the punctuation used to be fine and appropriate as “Ethics approval was granted by The Union Ethics Advisory Group of the International Union against Tuberculosis and Lung Diseases, Paris, France, IRB number EAG 53/18 and the Medical Research Council of Zimbabwe (MRCZ), IRB number MRCZ/A/2331”

Comment

xvii. Measurement of adverse events seems to have been recorded throughout treatment for respective patients. If so, this cannot be included as a potential risk factor in the predictive analysis as results would be prone to systematic bias. Those who remain in care long enough to develop and record ADRs will be different from those who do not. If ADRs are to be included, this should be recorded within a pre-defined period around baseline i.e., within 14 days after starting treatment for example. Thereafter, the analysis should be rerun.

Response

Thank you for the valid comment. We have removed SAEs from the multivariate regression model.

Comment

xviii. Table 4 – stepwise approach to model selection given in methods – but no p-values shown in table?

Response

Thank you for raising this comment. We did not include p-values as they would be redundant given that the reported univariate and multivariate-adjusted relative risks with 95% confidence intervals are more informative. When the 95%CI doesn’t cross the null value, then the factor is significant.

Comment

xix. How would the reader interpret significant risk factors for variables with categories that were ‘not recorded’? i.e., streptomycin DST pattern, Ethambutol DST pattern? Isoniazid DST pattern, missed doses?

Response 

We appreciate your comment, however the interpretation for unrecorded data will be largely not make programmatic interpretation especially for the “missed doses” variable. We however included all the patients with unrecorded data in the univariate and multivariate-adjusted regression models for the selected variables since excluding them will greatly diminish the eligible sample given the large amount of missing data as is often the case with routinely collected programme data.

Comment

3. Discussion

i. Line 316-324 – please rephrase language used here.

Response

Thank you. Rephrased to make the statement clearer as” The proportion with an unfavourable treatment outcome (death, Treatment failure, LTFU and Not evaluated) increased annually from 0% in 2010 to 45% in 2015 (Figure-1). 

Factors associated with an unfavourable outcome among patients on MDR-TB treatment are shown in Table 4. 

Those who were HIV-positive and ART-naive (ARR=2.83; 95% CI: 1.44-5.57) and those who missed >10% of their MDR-TB treatment dosses (ARR=2.41; 95% CI: 1.59-3.67) were more likely to have an unfavourable treatment outcome

Comment

ii. Line 325-326: what was the breakdown of patients across sites? Did some sites have a disproportionally higher number of patients?

Response 

 Thank you for the comment. Fifty-two out of sixty-three district hospitals that notified an MDR/RR TB patient during 2010-2015 contributed different numbers of the patients included in the study who were not refered to rural health centres. Thirty polyclinics from the two metro-politan provinces also contributed varied numbers but less patients per site as compared to what the district hospitals notified.

Across sites, yes the numbers were different depending on various factors as per the inclusion criteria described in the study design. 

Comment

iii. Please consider the use of language in the strength and limitations sections of the discussion. 

Response

The language used is deemed appropriate for a scientific paper as it arranges the strengths and limitations as first, second, third etc thus allowing the reader to follow the sequence of the issues highlighted.

Comment

iv. Line 328 – how were operational challenges assessed in this analysis?

Response

Since this study was based on a retrospective review of routine patient records. Therefore we did not conduct a process evaluation which would have been useful in determining operational challenges encountered in treatment and management of patients on MDR-TB treatment

Comment

v. Line 336 and 337 seems to be in contradiction to line 325-326

Response

Thank you for the observation, we however think there is no contradiction as a district has both urban and rural population. The participants included in our study were predominantly from the urban part of the district yet those that were excluded were predominantly from the rural part of the same district. It is not necessarily correct to assume that districts have only rural population

Comment

vi. Line 343-344 – if these changes were assessed during treatment, I’m not sure they would be suitable in your predictive analysis for the reasons of systematic bias as mentioned above

Response 

 Thank you. Comorbidities were determined during eliciting patient history at MDR-TB treatment initiation. We have revisited line 343-344 and specified this.

Comment

vii. Line 357-359 – could you comment on time on ART

Response

Thank you. unfortunately data on ART initiation start dates were not available in the DR-TB registers hence we could not assess timing of ART in relation to MDR-TB treatment commencement. We have added this as a limitation to paragraph of the discussion section.

Comment

viii. Line 363 – still unsure of exactly how missed doses was defined? Please could this be clarified in the methods.

Response

Thank you, Missed doses are clearly defined under Methods (Operational Definitions) Line 255 as percentage of the number of days with missed doses divided by the total number of days a patient was on treatment up until date of outcome. 

Comment

ix. Line 387 – at what time-point was cured/completed outcomes assessed? Was this by 24 months on treatment? As mentioned above – the time point at which outcomes were assessed is not clear here.

Response 

Thank you. Time point of cured/completed outcomes was assessed at the end of treatment duration. 

Comment

x. Line 411 – could you provide a measure of time between missed doses in the results section. 

Response 

Thank you. The following statement has been added to the results section under implications of study results: “ If a patient consecutively misses doses of two months or more, the patient is considered lost to follow-up and will restart treatment upon return”.

Comment

xi. Line 419-423 – it is still not clear how findings presented here link to the conclusions presented.

 Response

Our conclusions were linked to our major study findings which are as follows: 1) the MDR-TB treatment success rate was lower than the WHO treatment success target 2) treatment outcomes were better among those on ART whilst 3) missed doses of more than 10% were associated with poor outcomes. In addition, we have also made mention of the high occurrence of SAEs

We thank Reviewer 1, 2 and 3 including the Editor for a thorough review and hope we have satisfactorily addressed all the grey areas. We are more than pleased to attend to any further comment that may require a follow-up response

28 August 2019

PLOS ONE

Dear Editor in Chief

Please find below a reply letter to our submission, titled:

PONE-D-19-16150: Treatment outcomes of Multi drug resistant and Rifampicin resistant Tuberculosis in Zimbabwe: A cohort analysis of patients initiated on treatment during 2010 to 2015

We thank the reviewers and the Editor for their valuable comments to our initial submission. We have attempted to address all the queries made. We provide below a point-by-point response to the queries raised and we have also provided both the clean version of the manuscript and a document with track changes.

Editor comments

Comment #1

Response

Noted thank you. This has been addressed as per requirement

Comment #2

In ethics statement in the manuscript and in the online submission form, please provide additional information about the patient records/samples used in your retrospective study. Specifically, please ensure that you have discussed whether all data/samples were fully anonymized before you accessed them and/or whether the IRB or ethics committee waived the requirement for informed consent. If patients provided informed written consent to have data/samples from their medical records used in research, please include this information.

Response

Thank you for raising this very important ethical issue. Data was fully anonymised as only unique identifiers in the form of TB registration numbers were abstracted onto the data collection proforma in place of patient names. The Ethics committee waived the requirement for individual informed consent since this was a retrospective study and the Ministry of Health and Child Care had given permission to access patients clinical records at participating centres.

Comment #3

Please note that all PLOS journals ask authors to adhere to our policies for sharing of data and materials: https://journals.plos.org/plosone/s/data-availability. According to PLOS ONE’s Data Availability policy, we require that the minimal dataset underlying results reported in the submission must be made immediately and freely available at the time of publication. As such, please remove any instances of 'unpublished data' or 'data not shown' in your manuscript and replace these with either the relevant data (in the form of additional figures, tables or descriptive text, as appropriate), a citation to where the data can be found, or remove altogether any statements supported by data not presented in the manuscript

Response

Thank you for raising this. The data not shown on distribution and type of severe adverse events has now been added as Table 3.

Comment #4

 Our internal editors have looked over your manuscript and determined that it is within the scope of our Antimicrobial Resistance call for papers. This collection of papers is headed by a team of Guest Editors for PLOS ONE: Kathryn Holt (Monash University and London School of Hygiene and Tropical Medicine), Alison H. Holmes (Imperial College London), Alessandro Cassini (WHO Infection Prevention and Control Global Unit), Jaap A. Wagenaar (Utrecht University). The Collection will encompass a diverse range of research articles; additional information can be found on our announcement page: https://collections.plos.org/s/antimicrobial-resistance. If you would like your manuscript to be considered for this collection, please let us know in your cover letter and we will ensure that your paper is treated as if you were responding to this call. If you would prefer to remove your manuscript from collection consideration, please specify this in the cover letter.

Response

Thank you for considering our manuscript for the PLOS ONE collection on Antimicrobial Resistance. We would like to express our interest to have our manuscript considered as such and we give you explicit permission to process accordingly. We have indicated our interest in the attached cover letter

Comment #6

 We note that you have indicated that data from this study are available upon request. PLOS only allows data to be available upon request if there are legal or ethical restrictions on sharing data publicly. For information on unacceptable data access restrictions, please see http://journals.plos.org/plosone/s/data-availability#loc-unacceptable-data-access-restrictions.

Response

Thank you for the comment. Unfortunately we are not permitted to share our dataset due to our local ethics clearance committee and the Ministry of Health and Child Care who are the custodians of all data related to the National TB Programme. We have provided details of institutional heads for both organizations in the cover letter as requested.

Comment #7

We note that you have included the phrase “data not shown” in your manuscript. Unfortunately, this does not meet our data sharing requirements. PLOS does not permit references to inaccessible data. We require that authors provide all relevant data within the paper, Supporting Information files, or in an acceptable, public repository. Please add a citation to support this phrase or upload the data that corresponds with these findings to a stable repository (such as Figshare or Dryad) and provide and URLs, DOIs, or accession numbers that may be used to access these data. Or, if the data are not a core part of the research being presented in your study, we ask that you remove the phrase that refers to these data.

Response

The data that were not shown have now been added as “Table 3”.

Additional Editor Comments:

Comment #1

In addition to the Reviewer comments, I have the following comments that the authors should consider.

1. The authors should consider revising their concluding sentence which reads "There is a need for increased uptake of ART". HIV status and ART status was reported at initiation of treatment. Some of the patients who are HIV positive may be newly diagnosed and may not have initiated ART yet - it would be useful to look and see if those who were HIV positive but not on ART at treatment initiation, initiated ART between 2 weeks to 2 months depending on the CD4 count (as per guidelines). You may find some of the HIV positive patients initiated ART after they initiated DR-TB, and therefore it is difficult to make inferences about treatment outcomes.

Response

Thank you. In our study, dates on when ART was initiated in relation to commencement of TB treatment were not collected. However, the ART status of those who newly tested HIV-positive are supposed to be updated in the DR-TB register throughout their course of MDR-TB treatment period together with date of ART initiation although the information of ART start dates were missing in our case. As a result this limits our capabilities to determine impact of ART timing on DR-TB treatment outcomes, however we feel there is merit to conclude that ART does improve DR-TB treatment outcomes. Instead we have added to the abstract the following sentence: “Assessing timing of ART initiation in relation to TB treatment outcomes will also require future exploration.” We hope this response is satisfactory.

Comment #2

2. Line 120 - I find the use of "short-course" confusing here, since the WHO has guidelines for short- and long-course DR-TB treatment. I would consider rephrasing. The reference [2] "DOTS-Plus and Green Light Committee" refers to 6-8 months with first-line anti-TB treatment.

Response

Noted, the term “Short course” has been removed and the sentence now reads as follows: “In 2000, WHO recommended the 20-24 month standardized second-line drug (SLDs) regimens for treatment of MDR/RR-TB patients in resource-limited settings”. We have also removed the reference [2] on the DOTS-Plus and the Green Light Committee.

Comment #3

3. Line 134 - requires a reference for "WHO has recommended shorter regimens".

Response

Thank you. We have now referenced the 2019 WHO consolidated guidelines on drug-resistant tuberculosis treatment.

Comment #4

4. Line 150 - Please clarify what the short course regimen is, in your setting (e.g. shorter, injectable-based, 9-12 month regimen)

Response

Noted, thank you. Now clarified and it reads: “In 2016, the short treatment regimen which is over 9-11 months and injectable based was adopted”

Comment #4 (Duplication of numbering)

 Line 132 - The authors refer to the shorter regimen and Line 152 - authors state that the short regimen has been rolled out in the NTP. Since the paper focuses on the standard long-course RR/MDR-TB regimen (18-24 month), between 2010 - 2015, the authors need to mention why the paper is relevant (since short course has been adopted by the NTP).

Response

The short treatment course was piloted since June 2018 and is made available only in a few districts. Even in districts where short regimen is available, very few patients are started on it due to issues in assessing eligibility. Up to now most districts are still implementing the 18-24 month regimen. The short regimen might have been initiated in less than 5% patients. The study is relevant as it informs on the treatment outcomes with longer regimens and also provides a baseline estimate for tracking the successful outcomes rates with scale up of short regimen in future. 

Comment #5

5. Line 189 - write out "MTB/Rif" the first time it appears in the text; Line 192 - write out "PTB" the first time it appears in the text e.g. pulmonary TB (PTB); Line 224 - write out "PMDT" the first time it appears in the text; Line 230-233 - write out plus (+) and Lopinavir/Ritonavir and the correct abbreviation LPV/r

Response

Line 189-"MTB/Rif" now appears in full the first time it appears in the text

Line 192-“PTB” now appears as “Pulmonary TB” the first time it appears in the text

Line 224-“PMDT" written in full the first time it appeared in line 149.

Line 230-233- Lopinavir/Ritonavir) (ABC + 3TC + EFV now reflected and LPV/r now reflected

Comment #6

6. The authors have introduced selection bias, which is a limitation of the study. The study includes MDR/RR-TB patients who initiated treatment between 2010 and 2015 and continued treatment at either district hospitals or urban polyclinics (Line 237). Those who were referred to primary health facilities were excluded (Line 239). This may contribute to why treatment outcomes are different to what is reported to the WHO by the NTP of Zimbabwe. Those who are referred to primary health facilities are considered more stable, tolerating the regimen, and live close to the PHC. These factors, including distance, have been shown to be associated with treatment outcomes, adherence to treatment and utilization of treatment services. The study population therefore presents patients who were unstable/not tolerating the regimen and/or those who lived >10km from the health facility. The fact that the study does not include any treatment outcomes for patients referred to primary health facilities is a limitation.

Response

Thank you very much for concurring with the authors that the study had some limitations as referred above. The above Editor concern has been articulated at length as a study limitation and authors acknowledged this quite extensively

Comment #7

7. Line 219 - please clarify what constitutes "stable". What criteria is used to determine if a patient is stable?

Response

“Stable” is a term used to describe those patients who were able to ingest medication, did not show signs of adverse drug reaction and had all the laboratory investigations within normal limit. This description now appears in line 219

Comment #8

8. Line 252 - Please clarify how "missed doses" is reported. Is this patient self-report or obtained from clinical data (e.g. dispensed)?

Response

“Missed doses” are determined from patient clinical records specifically from the patient treatment card. As the directly observed treatment is provided, the ‘right tick’ is made against the particular date in the treatment card by the clinician when drugs are consumed. In case if the patient fails to report for DOT at the health facility, a follow up is done and in the event the patient is not located and misses his drugs for that day, it is marked as missed with ‘wrong’ tick against that date. We considered all the ‘wrong’ tick till the date of treatment outcome as ‘Missed doses’, Later we calculated the percentage of missed doses per total number of days the patient was receiving the treatment.

Comment #9

9(1) Line 269 - regarding treatment outcomes; (1) authors should refer to the WHO reporting framework and include the definitions of the outcomes.

Response

Noted. “Table of Treatment Outcomes definition” referred as Fig 2

9(2) authors needs to include when the outcome was defined (i.e. how long were patients followed-up for, what is the person-time?), (3) the definition of primary outcome seems unusual - can authors include a reference for this?, (4) since death and LTFU are assigned when the outcome occurs, it is more appropriate to combine death + LTFU in your outcome definition and use Cox Proportional Hazard regression

Response

Patient TB treatment outcomes were defined at 24 months from date of each patients MDR-TB treatment start date until date of their respective outcomes for deaths, LTFUs, treatment failure and “outcome not evaluated”. Those classified as LTFU were defined as a patient whose MDR-TB treatment was interrupted for two or more consecutive months for any reason as per WHO guidelines. Those cured or who completed their treatment were categorized as successful treatment outcomes, whereas the others were categorized as unsuccessful treatment outcomes in line with recent meta-analysis studies by Kibret et al and Johston JC et al. We have added this to the data analysis section of our paper

The 473 patients in our study contributed 672.5 person-years. We have added this to the first paragraph of the results section. Thank you for suggesting that we use Cox Proportional Hazards regression. We considered this at length as the most appropriate analysis method, however we finally abandoned it because there were a number of patients who had missing dates on treatment outcomes hence this would lower our sample size.

Comment #10

10. Line 280 - 282 - IRB protocol numbers need to be included. Line 283 - "health and child care" should be corrected as in Line 282 "Ministry of Health and Child Care"

Response

Line 280 - 282 - IRB protocol numbers now included.

Line 283 - "health and child care"- Both sentences line 282 and 283 read as “Ministry of Health and Child Care” from original texts

Comment #11

Line 233 - Guidelines for the use of cotrimoxazole (CPT) should be included in the Methods. 

Response

A reference for the guidelines of use of Cotrimoxazole Preventive Therapy is now indicated and the reference section has been updated

Comment #12

12. Line 569 - some key variables are missing in Table 2. Examples include resistance pattern (e.g. MDR-TB, RR-TB (mono; isoniazid sensitive) or RR-TB with additional resistance unknown), time on ART, CD4 count, smear microscopy results, weight/BMI (Line 268), EPTB vs. PTB.

Response

Thank you for observing that some key variables are missing in Table 2. All the above are missing data as explained in the section on study limitations

Comment #13

13. Line 236 and Table 1 - please clarify if the study includes children? From Table 1 n=24 were <24 years. It is not clear if this is 18-24 or 0-24? If the latter, then this should be further categorized as children (<10) or adolescents (10-24) or further as young adolescent (10-14), older adolescent (15-19) and young adult (20-24).

Response

The study does include children. We had collapsed the age groups <24 years because of small numbers, however we have reverted back to the disaggregation of <5, 5-14 and 15-24 years in Table 1 but however maintained the combined age group of <24 years in the multivariate regression analysis table so as to allow for adequate numbers for calculating relative risks and their 95% confidence intervals. We also used the age categories <24, 25-34, 35-44, 45-54 and 55+ years as these are the standard age groups for reporting TB data to WHO and are also used in several studies to allow for comparisons.

Comment #14

14. Table 2 Encountered SAEs - Is this during treatment or at treatment initiation. If the former, then this should be separated out from the characteristics at treatment initiation (Clarify that Table 2 includes clinical characteristics AT treatment initiation). Would be useful to include/mention the most common SAEs reported in the Table (i.e. Table 3?)

Response

Thank you for the point. The encountered SAEs were during treatment and we have rephrased the variable as “Encountered SAE during treatment”. Seeing that some patients may have also initiated ART during MDR-TB treatment and that we have the variable on missed treatment dosses in this table, we have instead rephrased the table title as “Clinical characteristics of MDR/RR-TB patients at baseline and/or during treatment among MDR/RR-TB patients initiated on treatment in Zimbabwe, 2010-2015”. We hope that this is satisfactory.

Comment #15

15. Table 3 Outcomes should add up to 100% - Failed should be 0.9% and not <1 and LTFU 8.2% (39/473). Therefore, 26.4 + 8.2 + 0.9 + 3.4 = 100%

Response

Many thanks for pointing this out. We have replaced the percentage for those with treatment failure with 0.8% however we note that the percentages for the different end-of-TB treatment outcomes add up to 99.9% due to rounding-off errors. We have specified this as a footnote to the table.

Comment #16

16. Table 4 - consider defining a resistance pattern variable - which should encompass the diagnostic test and the DST results.

Response

Thank you for the comment. This information was not available due to missing data on CDST results which the authors highlighted as a study limitation in the discussion section.

Comment #17

17. Table 4 – The current model ignores the time-varying nature of adherence.

Response

This is indeed a valid comment and we ought to have included missed doses as a time dependent variable in our univariate and multivariate-adjusted regression models. However, we did not have data on the specific time points when MDR-TB treatment doses were missed as these were aggregated for each patient by the time they achieved their treatment outcome

Comment #18

18. Figure 1 - It would be useful to present outcomes as typically reported by the WHO (e.g. see https://www.who.int/tb/publications/global_report/gtbr2017_main_text.pdf - Page 80)

Response

Thank you for the comment. The term “Unfavourable Outcome” has been further clarified in the text as per WHO reporting system (Death, Treatment failure, LTFU and Not Evaluated)

 Reviewer #1: 

I appreciate the work of the Authors to try to define factors associated with unfavourable outcomes in RR and MDR TB in a Country with a very high percentage of HIV co-infected patients.

I suggest the Authors to stress the need for individualized treatment regimens in order to decrease side effects (BMC Infect Dis. 2019 Jun 28;19(1):564. doi: 10.1186/s12879-019-4211-0.) and to cite the need for better treatment compliance (BMC Infect Dis. 2017 Jan 21;17(1):91. doi: 10.1186/s12879-017-2200-8. and Multidiscip Respir Med. 2018 Nov 9;13:41. doi: 10.1186/s40248-018-0154-3. eCollection 2018)

Response

The authors appreciate this valid comment. On study implications, line 409, the authors stresses the need for NTP to adopt shorter treatment regimen for MDR-TB treatment which has been shown to have fewer SAEs. A reference has been added and cited in the reference section as advised

Line 417, the authors stresses the need to ensure that patients do not have missed doses during MDR-TB treatment in order to lessen their risk of having unfavourable outcomes. . A reference has been added and cited in the reference section as advised.

 Reviewer #2: 

TITLE – Consider adding ‘Urban Setting’ to the title since the population belongs to urban areas and probably a bias to favorable outcome.

Response

Thank you for raising this important point, however the study was not primarily set for the urban population as some district hospitals serve mostly rural populations thus the study population was both urban and rural

Abstract – appropriate – consider elaborating the methods.

Response

Thank you for this comment. The methods section of the abstract has been elaborated as advised. The following statement has been added “A generalized linear model with a log-link and binomial distribution or a poisson distribution with robust error variances were used assess factors associated with “unfavorable outcome”. The unadjusted and adjusted relative risks were calculated as measure of association. A �value< 0.05 was considered statistically significant”.

Introduction – Describes the problem in a complete and relevant manner. Aim of the study is well reported.

Response

Thank you, well appreciated

Comment

Method – Please consider mentioning the type of phenotypic cultures utilized.

Response

Thank you. The type of phenotypic cultures utilized is now indicated as BBLTM MGITTM Mycobacterial Growth Indicator Tubes (Becton Dickinson, Sparks, MD

Comment

Operational definition of treatment outcomes, although standardized as per WHO – please reiterate here, this can prevent a bit of confusion

Response

 Noted, Fig 2 has been added to clarify

Comment

Results – Now as mentioned by the authors in terms of limitation – the absence of data on management of SAE, CDST results, CD4 count and comorbidity data. I can suggest certain key data variables which are missing, if available, consider providing them.

BMI, radiological evidence of cavities as they may have direct effect on outcome as well as occupational history and smoking status of the cohort, both of which are strong risk factors for developing TB and unfavorable outcome

Response

We thank you for raising this very valid comment. As already mentioned as study limitations, the data suggested was missing and cannot be included

Comment

It is unfortunate that DST results of fluoroquinolones and second line injectables are not available, both of which are known to be risk factors for adverse outcome, if for the ~53 cases this data can be made available and analysed as a risk factor, it will greatly strengthen your study. It is a strong point for advocating the need of DST.

Response

Thank you for the comment. Of all the second-line injectables and flouroquinolones in use (i.e. kanamycin, Levofloxacin, Moxifloxacin and Cycloserine and Capriomycin alone as an alternative to those intolerant to kanamycin), only 3 patients were resistant to any one of the above hence we could not include this variable in the univariate and multivariate regression analysis because of the small numbers.

Comment

Data for Time to unfavourable outcome – is a strong variable that can help us in understanding the severity of the cohort population and help in future intervention. Did any SAE led to unfavourable outcome? The individuals who died or lost to follow up – were recorded to be failing the regimen?

Response

Thank you for raising this very important observation. In our univariate and multivariate regression encountering an SAE was not significant predictor of an unfavourable outcome as shown in Table 5. Also, due to deficiencies in our data and this being a retrospective study, we failed to assess the cause-effect relationship of SAEs and unfavourable outcome

Comment

Details of ART initiation is not provided neither the status of ART success or failing is analysed.

Response

Thank you. Unfortunately data on virological, immunological or clinical failure of patients initiated on ART were not available in the DR-TB registers hence required matching patient names to the ART registers given that there was no unique identifier linking the ART and MDR-TB programmes. However we do acknowledge the importance of this data and is something that was worth exploring and should be explored in future studies.

Comment

Discussion:

The authors have done proper analysis of their results, aptly described the limitation and strengths of the study as well as compared the results with similar studies.

Response

Thank you, comment well appreciated

Comment

Please elaborate on the reason for 180 people with no recorded time for culture conversion – they failed to culture convert or some of them died before or some could not give the sample

Response

Thank you for asking us to elaborate. These patients either did not have their samples examined because of contamination, spillages or did not have results (CDST) sent back to the health facility and the reasons are well explained under the section on study implications where the authors mentioned that the high proportion of patients who did not have CDST results during their treatment is cause for concern as this is essential in monitoring bacteriological response to treatment. A recent study from Zimbabwe showed leakages in receipt of sputum samples at NRLs, culture contamination among received sputum specimens leading to a reduced proportion of samples with CDST results and this was mentioned in the relevant section. The authors concluded by stressing that this CDST system will require improvements including feedback of CDST results to facilities in order to inform patient management.

Comment

How many patient had their sputum culture status reverted back to positive

Response

Thank you for this question. None had their sputum culture status reverted back to positive

Comment

Interestingly, ~14% of the study cohort was initiated on ATT beyond 30 days of RRTB diagnosis, was not found to affect the outcome. You may consider highlighting that delay in initiation for SLD.

Response

Thank you for pointing this out. We have mentioned the following in the discussion section: “Last, a significant number of patients delayed initiation of MDR-TB treatment by more than 30 days after RR-TB diagnosis. Whilst this was not associated with having an unfavourable outcome, this has got dire consequences at an individual level towards disease progression and at a population-level towards MDR-TB transmission in the community.”

 Reviewer #3:

Comment #1

Background is unclear and does not adequately contextualize the aims of the manuscript.

Response

Thank you for the comment. This study aimed at assessing the profile, treatment outcomes and factors associated with unfavourable treatment outcomes among patients initiated on MDR/RR-TB treatment. In view of this we started by highlighting the global burden of MDR/RR-TB and also mentioned the WHO MDR/RR-TB treatment targets as a benchmark to assess our poor performance in this regard. We further summarised some of the literature of risk factors associated with poor MDR-TB treatment outcomes before we highlighted the paucity of data on treatment outcome determinants in our setting. This is of interest to the Zimbabwe NTP in order to have tailored interventions aimed at improving our overall MDR-TB treatment success rate. We do appreciate your comment and would like your further guidance on specific sections that require rewriting in order to make the introduction more appealing.

Comment #2

Methods lack clarity on how outcomes were defined, time-points at which outcomes were assessed by, measurement of covariates like adverse events is not suitable, inferential analytic techniques are also questionable.

Response

Thank you for this comment. We have addressed your comment on outcome definitions and time-points for assessment of these outcomes are addressed in the data entry and analysis section in line with other reviewers comments. As for inferential analytical techniques, we would have preferred to use Cox Hazards regression analysis to generate hazard ratios, however due to a large proportion of missing data on time of outcomes, we resorted to generating relative risk instead. Also, as our outcome of interest in this manuscript is ‘unfavourable outcome’, which includes ‘death’, LTFU, failure and ‘not evaluated’ we considered binomial regression. If at all we were modelling for death or LTFU than the Cox Hazard regression would have been the only option.

Comment #3

Results are sparse and could be described in greater detail.

Response

Thank you. In our analysis, we generated tables that responded to each of our set specific objectives and were hesitant to delve into greater detail for fear of digressing from our set objectives. However, we would appreciate if you can specify particular detail which would be of interest to the readers of our paper and is also aligned with our study objectives.

Comment #4

It is not very clear how background, aims, methods, results tie into the discussion and conclusions. 

Response

Thank you for the comment. We are however surprised that this is the case as your comment is contradictory to the other 2 reviewers. In writing the paper we carefully attempted to logically align our discussion to the results presented which in turn were informed by the background, aims and methods of the study. We hope this is understandable with you

Comments:

1. Keywords 

a. Repeated (full-text and abbreviations used for the same word)

Response

We have gone through the whole manuscript and ensured that full-text and abbreviations are reported together only once, following which only abbreviations are mentioned were necessary.

a) Background 

Comment

Tuberculosis to be written out in full before abbreviated

Response

Thank you. Tuberculosis now written out in full the first time it is used before being abbreviated

Comment

Incorrect order of abbreviations for RR/MDR-TB. Authors write this out in full starting with MDR-TB.

Response

Thank you for this. We appreciate that MDR/RR-TB can be written as RR/MDR-TB however after careful consideration, the study investigator team opted to align with MDR/RR-TB which is used in the WHO global TB reports and guidelines. We hope this is understandable.

Comment

Please rephrase last sentence. This is unclear

Response

Thank you for the comment. The last sentence on the background now rephrased and reads “The profile, management, and factors associated with unfavourable treatment outcomes of MDR/RR TB have not been systematically evaluated in Zimbabwe”.

Comment

b) Design

1) Please provide more detail regarding the methods you employ to assess treatment outcomes and risk factors for such outcomes. Study sites, participants, data collected, analytic methods used, definition of outcomes assessed, time point outcomes were assessed by etc.

Response

Study sites were all district hospitals and urban polyclinics in Zimbabwe. The study participants were all MDR/RR-TB patients initiated on treatment between 2010-2015 under the Zimbabwe NTP and continued their treatment in the above mentioned health facilities. This information is stated in the section on “Study Population”. We have further on added the total number of district hospitals and urban polyclinics which totalled eighty-two under study design. As per your earlier request we have specified the time point outcomes assessed in the “Data entry and analysis” section.

2) How were participants selected for the final analysis?

Response

We thank you for asking us to explain. Of the total 935 MDR/RR-TB patients initiated on treated during study reference period , the 473 (51%) of patients who met the study inclusion criteria of those started on MDR/RR-TB treatment and continued their 20-24 month treatment at district and urban polyclinics were all included in the study

Comment

c) Results 

i. How was missing MDR-TB treatment doses defined? Please add to methods.

Response

Thank you. Percentage of missed doses is defined as percentage of the number of days with missed doses divided by the total number of days a patient was on treatment up until date of outcome. This has been added to methods section

Comments

ii. Did outcomes differ between RR-TB and MDR-TB?

Response

Thank you for this valid comment. As shown in Table 5, based on the individual analysis of isoniazid, streptomycin and ethambutol resistant patterns at baseline, there were no differences in outcomes between RR-TB and MDR-TB patients. Furthermore, it is difficult to conclusively distinguish between RR-TB and MDR-TB patients as the majority of patients did not have DST results recorded or at least for all second-line TB drugs..

Comment

iii. How were covariates selected to be included in the main effects model? Please add to methods.

 Response

 Thank you for the comment. All covariates with a p<0.25 and those that are biologically plausible predictors of unfavourable outcomes were added to the multivariate regression model. This is mentioned in the “Data entry and analysis” section.

Comment

iv. Why was ‘not evaluated’ outcome considered unfavourable?

Response 

Those “not evaluated” were included in those with unfavourable outcomes as stated in the “Data entry and analysis” section in line with recent meta-analysis studies by Kibret et al and Johston JC et al. The WHO in its annual TB report, considers ‘not evaluated’ as ‘unfavourable’ or ‘unsuccessful’ outcome

Comment

d)Conclusion

v. Stating that outcomes were poor coupled with high occurrence of adverse events make it seems like this was a significant risk factor – but it wasn’t?

Response 

Thank you for the comment. Whilst we acknowledge that adverse events were not a significant risk factor, in paragraph 6 of the discussion section we only highlight that the high proportion of SAEs in this patient cohort (i.e. one in four patients) is a cause for concern and do not insinuate that this is a risk factor for unfavourable outcomes. We further advocate for more standardized recording and reporting of SAEs for better tracking by the NTP and also advocate for wider adoption of the shorter MDR-TB treatment regimen

Comment

vi. Conclusions and recommendations are based on adverse events; however this is not mentioned as a significant predictor in the results. Therefore, how are these conclusions relevant to your findings?

Response 

Thank you. Please refer to the earlier response above.

2. Introduction -Comment

i. Capitalize ‘tuberculosis’ line 112

Response 

“T” Now in capital letters

Comment 

ii. Don’t start a sentence with an abbreviation – line 114 and other places within the text

Response 

 Line 114 and 118 corrected

Comment 

Insert “The” before World Health Organisation

Response

The World Health Organization’ – line 116

“The” Inserted

Comment

iii. Line 120 – is this reference from 2000?

Response

Thank you. Yes, we can confirm that the reference is from 2000

Comment

iv. Line 128-129 – please clarify what the second part of this sentence means

Response

The authors of the systematic review paper stated that studies included in the review were largely from countries with low HIV coinfection. They then recommended for systematic assessment of treatment outcomes in high HIV coinfection countries of sub-Saharan Africa

Comment

v. Could you comment on the incidence of ADRs on second-line TB treatment? Line 130-134

Response

Thank you for raising this comment. We could not generate incidence rate of ADRs as we did not have data on time of their occurrence.

Comment

vi. Could you comment on eligibility for newer shorter regimens? Line 130-134

Response

Thank you for asking us to comment. Eligibility for newer shorter regimens is when there is no risk or confirmed resistance to any second line medicines or any medicine used in the shorter treatment regimen

Comment

vii. However, unsure of commenting on newer short course regimens – what does this have to do with assessing treatment outcomes in your traditional-long course cohort? Shorter regimens may be more pertinent in your discussion.

Response

Our apologies as we do not clearly understand your comment. If we understand well, you suggest that we mention the importance of the newer short course regimen as an alternative to the traditional long course regimen. We have touched on this in paragraph 10 of the discussion section were we suggest that the NTP should adopt wider scale up of the shorter treatment regimen which has fewer SAEs although treatment outcomes are comparable to the long MDR-TB treatment regimen.

3. Methods

Comment 

viii. Could a figure describing the general setting be included to better understand the public healthcare setting? And thereafter selection process 

Response 

Thank you for suggesting this. We have attempted to present this information graphically but have concluded that the text description mentioned in the General setting and the description of diagnosis and treatment of MDR/RR-TB gives an overall description of Zimbabwe’s public health setting in the context of the National TB Programme. We are happy to clarify any specific sections that are unclear to you.

Comment

ix. Line 212-213 – WHO recommended DOTS-plus during the study period? Or as above in 2000?

Response

In 2000, Tense changed to reflect “Was” instead of “be”

Comment

x. Line 237 -238 please rephrase

Response

Rephrased as per reviewer 3 comments, sentence now short and more precise

Comment

xi. Would patients referred for DOTS-plus who were then excluded from the analysis be systematically different to those who were included?

Response

 Yes we do acknowledge this possibility given that the treatment success rate for our study population was better than that reported by the NTP for all patients due to their high proportion of unevaluated patients. Furthermore in rural settings health facilities are less accessible due to long distances from households and there is less specialized care in rural facilities. We have stated this as a limitation in the discussion section.

Comment

xii. Line 243 – consistency with capitalization

Response

We have noted this and addressed it accordingly.

Comment

xiii. Data quality between sites could differ? Were there any systematic differences between quality (missing data) etc. between sites?

Response 

Thank you for this comment, however there were no systematic differences between sites in terms of data quality. Missing data was across all sites because the tools used by all sites were the same and no data was collected on those variables. Also all the sites sent their samples to the same reference laboratory which did not give results back on CDST. Furthermore 10%of the data was subjected to quality assurance and originally missing data was completed before final data entry. 

Comment

xiv. Since the primary outcome of interest is death and LTFU which can occur any time from treatment initiation – would you consider using a Cox Hazard Model to determine hazard ratio estimates of ‘unfavourable outcomes’? Considering time to outcome may be an important factor which could be affected differently by different risk factors. Furthermore, was LTFU and death separated and did risk factors differ by these outcomes? Lastly, what is the motivation for including ‘not evaluated’ as an ‘unfavourable outcome’?

Response

 As mentioned above we could not perform Cox Hazard regression analysis due to the high proportion of missingness on time to event data. We also reserved further comparisons on any differences between predictors of mortality and LTFU for another paper as we felt this was too much information for one paper. We hope reviewers understand our position.

Comment

xv. Please include a time-point at which outcomes were assessed by. Did everyone have a minimum follow-up etc.?

Response

 Yes, since this was a retrospective cohort study based on routinely collected programme data, all patients were allowed a follow-up period for their potential 20-24 month MDR-TB treatment duration.

Comment

3Results 

xvi. Line 287 – punctuation

Response 

Thank you, we consider the punctuation used to be fine and appropriate as “Ethics approval was granted by The Union Ethics Advisory Group of the International Union against Tuberculosis and Lung Diseases, Paris, France, IRB number EAG 53/18 and the Medical Research Council of Zimbabwe (MRCZ), IRB number MRCZ/A/2331”

Comment

xvii. Measurement of adverse events seems to have been recorded throughout treatment for respective patients. If so, this cannot be included as a potential risk factor in the predictive analysis as results would be prone to systematic bias. Those who remain in care long enough to develop and record ADRs will be different from those who do not. If ADRs are to be included, this should be recorded within a pre-defined period around baseline i.e., within 14 days after starting treatment for example. Thereafter, the analysis should be rerun.

Response

Thank you for the valid comment. We have removed SAEs from the multivariate regression model.

Comment

xviii. Table 4 – stepwise approach to model selection given in methods – but no p-values shown in table?

Response

Thank you for raising this comment. We did not include p-values as they would be redundant given that the reported univariate and multivariate-adjusted relative risks with 95% confidence intervals are more informative. When the 95%CI doesn’t cross the null value, then the factor is significant.

Comment

xix. How would the reader interpret significant risk factors for variables with categories that were ‘not recorded’? i.e., streptomycin DST pattern, Ethambutol DST pattern? Isoniazid DST pattern, missed doses?

Response 

We appreciate your comment, however the interpretation for unrecorded data will be largely not make programmatic interpretation especially for the “missed doses” variable. We however included all the patients with unrecorded data in the univariate and multivariate-adjusted regression models for the selected variables since excluding them will greatly diminish the eligible sample given the large amount of missing data as is often the case with routinely collected programme data.

Comment

3. Discussion

i. Line 316-324 – please rephrase language used here.

Response

Thank you. Rephrased to make the statement clearer as” The proportion with an unfavourable treatment outcome (death, Treatment failure, LTFU and Not evaluated) increased annually from 0% in 2010 to 45% in 2015 (Figure-1). 

Factors associated with an unfavourable outcome among patients on MDR-TB treatment are shown in Table 4. 

Those who were HIV-positive and ART-naive (ARR=2.83; 95% CI: 1.44-5.57) and those who missed >10% of their MDR-TB treatment dosses (ARR=2.41; 95% CI: 1.59-3.67) were more likely to have an unfavourable treatment outcome

Comment

ii. Line 325-326: what was the breakdown of patients across sites? Did some sites have a disproportionally higher number of patients?

Response 

 Thank you for the comment. Fifty-two out of sixty-three district hospitals that notified an MDR/RR TB patient during 2010-2015 contributed different numbers of the patients included in the study who were not refered to rural health centres. Thirty polyclinics from the two metro-politan provinces also contributed varied numbers but less patients per site as compared to what the district hospitals notified.

Across sites, yes the numbers were different depending on various factors as per the inclusion criteria described in the study design. 

Comment

iii. Please consider the use of language in the strength and limitations sections of the discussion. 

Response

The language used is deemed appropriate for a scientific paper as it arranges the strengths and limitations as first, second, third etc thus allowing the reader to follow the sequence of the issues highlighted.

Comment

iv. Line 328 – how were operational challenges assessed in this analysis?

Response

Since this study was based on a retrospective review of routine patient records. Therefore we did not conduct a process evaluation which would have been useful in determining operational challenges encountered in treatment and management of patients on MDR-TB treatment

Comment

v. Line 336 and 337 seems to be in contradiction to line 325-326

Response

Thank you for the observation, we however think there is no contradiction as a district has both urban and rural population. The participants included in our study were predominantly from the urban part of the district yet those that were excluded were predominantly from the rural part of the same district. It is not necessarily correct to assume that districts have only rural population

Comment

vi. Line 343-344 – if these changes were assessed during treatment, I’m not sure they would be suitable in your predictive analysis for the reasons of systematic bias as mentioned above

Response 

 Thank you. Comorbidities were determined during eliciting patient history at MDR-TB treatment initiation. We have revisited line 343-344 and specified this.

Comment

vii. Line 357-359 – could you comment on time on ART

Response

Thank you. unfortunately data on ART initiation start dates were not available in the DR-TB registers hence we could not assess timing of ART in relation to MDR-TB treatment commencement. We have added this as a limitation to paragraph of the discussion section.

Comment

viii. Line 363 – still unsure of exactly how missed doses was defined? Please could this be clarified in the methods.

Response

Thank you, Missed doses are clearly defined under Methods (Operational Definitions) Line 255 as percentage of the number of days with missed doses divided by the total number of days a patient was on treatment up until date of outcome. 

Comment

ix. Line 387 – at what time-point was cured/completed outcomes assessed? Was this by 24 months on treatment? As mentioned above – the time point at which outcomes were assessed is not clear here.

Response 

Thank you. Time point of cured/completed outcomes was assessed at the end of treatment duration. 

Comment

x. Line 411 – could you provide a measure of time between missed doses in the results section. 

Response 

Thank you. The following statement has been added to the results section under implications of study results: “ If a patient consecutively misses doses of two months or more, the patient is considered lost to follow-up and will restart treatment upon return”.

Comment

xi. Line 419-423 – it is still not clear how findings presented here link to the conclusions presented.

 Response

Our conclusions were linked to our major study findings which are as follows: 1) the MDR-TB treatment success rate was lower than the WHO treatment success target 2) treatment outcomes were better among those on ART whilst 3) missed doses of more than 10% were associated with poor outcomes. In addition, we have also made mention of the high occurrence of SAEs

We thank Reviewer 1, 2 and 3 including the Editor for a thorough review and hope we have satisfactorily addressed all the grey areas. We are more than pleased to attend to any further comment that may require a follow-up response

---

## [Decision Letter · Decision Letter 1]

12 Dec 2019

PONE-D-19-16150R1

Treatment outcomes of Multi drug resistant and Rifampicin resistant Tuberculosis in Zimbabwe: A cohort analysis of patients initiated on treatment during 2010 to 2015

PLOS ONE

Dear Mr Matambo,

Thank you for submitting your manuscript to PLOS ONE. After careful consideration, we feel that it has merit but does not fully meet PLOS ONE’s publication criteria as it currently stands. Therefore, we invite you to submit a revised version of the manuscript that addresses the points raised during the review process.

We would appreciate receiving your revised manuscript by Jan 26 2020 11:59PM. To enhance the reproducibility of your results, we recommend that if applicable you deposit your laboratory protocols in protocols.io, where a protocol can be assigned its own identifier (DOI) such that it can be cited independently in the future. For instructions see: http://journals.plos.org/plosone/s/submission-guidelines#loc-laboratory-protocols

We look forward to receiving your revised manuscript.

Kind regards,

Denise Evans, PhD

Academic Editor

PLOS ONE

Reviewers' comments:

Reviewer's Responses to Questions

**Comments to the Author**

1. If the authors have adequately addressed your comments raised in a previous round of review and you feel that this manuscript is now acceptable for publication, you may indicate that here to bypass the “Comments to the Author” section, enter your conflict of interest statement in the “Confidential to Editor” section, and submit your "Accept" recommendation.

Reviewer #1: All comments have been addressed

Reviewer #2: (No Response)

Reviewer #3: (No Response)

2. Is the manuscript technically sound, and do the data support the conclusions?

Reviewer #1: Yes

Reviewer #2: Yes

Reviewer #3: No

3. Has the statistical analysis been performed appropriately and rigorously? 

Reviewer #1: N/A

Reviewer #2: Yes

Reviewer #3: No

4. Have the authors made all data underlying the findings in their manuscript fully available?

Reviewer #1: Yes

Reviewer #2: Yes

Reviewer #3: Yes

5. Is the manuscript presented in an intelligible fashion and written in standard English?

Reviewer #1: Yes

Reviewer #2: Yes

Reviewer #3: No

6. Review Comments to the Author

Reviewer #1: (No Response)

Reviewer #2: TITLE – Consider adding ‘Urban Setting’ to the title since the population belongs to

urban areas and probably a bias to favorable outcome.

Response

Thank you for raising this important point, however the study was not primarily set for

the urban population as some district hospitals serve mostly rural populations thus the

study population was both urban and rural

Reviewer’s Response: Author’s response acceptable.

Abstract – appropriate – consider elaborating the methods.

Response

Thank you for this comment. The methods section of the abstract has been elaborated

as advised. The following statement has been added “A generalized linear model with

a log-link and binomial distribution or a poisson distribution with robust error variances

were used assess factors associated with “unfavorable outcome”. The unadjusted and

adjusted relative risks were calculated as measure of association. A value< 0.05 was

considered statistically significant”.

Reviewer’s Response: Author’s response acceptable.

Introduction – Describes the problem in a complete and relevant manner. Aim of the

study is well reported.

Response

Thank you, well appreciated

Comment

Method – Please consider mentioning the type of phenotypic cultures utilized.

Response

Thank you. The type of phenotypic cultures utilized is now indicated as BBLTM

MGITTM Mycobacterial Growth Indicator Tubes (Becton Dickinson, Sparks, MD

Reviewer’s Response: Addition of information is appreciated.

Comment

Operational definition of treatment outcomes, although standardized as per WHO –

please reiterate here, this can prevent a bit of confusion

Response

Noted, Fig 2 has been added to clarify

Reviewer’s Response: Appropriate action taken.

Comment

Results – Now as mentioned by the authors in terms of limitation – the absence of data

on management of SAE, CDST results, CD4 count and comorbidity data. I can suggest

certain key data variables which are missing, if available, consider providing them.

BMI, radiological evidence of cavities as they may have direct effect on outcome as

well as occupational history and smoking status of the cohort, both of which are strong

risk factors for developing TB and unfavorable outcome

Response

We thank you for raising this very valid comment. As already mentioned as study

limitations, the data suggested was missing and cannot be included

Reviewer’s Response: Consider adding – lack of radiological data in the heading of limitation.

Comment

It is unfortunate that DST results of fluoroquinolones and second line injectables are

not available, both of which are known to be risk factors for adverse outcome, if for the

~53 cases this data can be made available and analysed as a risk factor, it will greatly

strengthen your study. It is a strong point for advocating the need of DST.

Response

Thank you for the comment. Of all the second-line injectables and flouroquinolones in

use (i.e. kanamycin, Levofloxacin, Moxifloxacin and Cycloserine and Capriomycin

alone as an alternative to those intolerant to kanamycin), only 3 patients were resistant

to any one of the above hence we could not include this variable in the univariate and

multivariate regression analysis because of the small numbers.

Reviewer’s Response: Author’s response accepted.

Comment

Data for Time to unfavourable outcome – is a strong variable that can help us in

understanding the severity of the cohort population and help in future intervention. Did

any SAE led to unfavourable outcome? The individuals who died or lost to follow up –

were recorded to be failing the regimen?

Response

Thank you for raising this very important observation. In our univariate and multivariate

regression encountering an SAE was not significant predictor of an unfavourable

outcome as shown in Table 5. Also, due to deficiencies in our data and this being a

retrospective study, we failed to assess the cause-effect relationship of SAEs and

unfavourable outcome

Reviewer’s Repsone: Clarification needed for patients – Lost to follow up or died- were the failing on the current regimen.

Comment

Details of ART initiation is not provided neither the status of ART success or failing is

analysed.

Response

Thank you. Unfortunately data on virological, immunological or clinical failure of

patients initiated on ART were not available in the DR-TB registers hence required

matching patient names to the ART registers given that there was no unique identifier

linking the ART and MDR-TB programmes. However we do acknowledge the

importance of this data and is something that was worth exploring and should be

explored in future studies.

Reviewer’s Repsonse: Please consider mentioning this shortcoming in the Discussion.

Comment

Discussion:

The authors have done proper analysis of their results, aptly described the limitation

and strengths of the study as well as compared the results with similar studies.

Response

Thank you, comment well appreciated

Reviewer’s Response: No further action needed

Comment

Please elaborate on the reason for 180 people with no recorded time for culture

conversion – they failed to culture convert or some of them died before or some could

not give the sample

Response

Thank you for asking us to elaborate. These patients either did not have their samples

examined because of contamination, spillages or did not have results (CDST) sent

back to the health facility and the reasons are well explained under the section on

study implications where the authors mentioned that the high proportion of patients

who did not have CDST results during their treatment is cause for concern as this is

essential in monitoring bacteriological response to treatment. A recent study from

Zimbabwe showed leakages in receipt of sputum samples at NRLs, culture

contamination among received sputum specimens leading to a reduced proportion of

samples with CDST results and this was mentioned in the relevant section. The

authors concluded by stressing that this CDST system will require improvements

including feedback of CDST results to facilities in order to inform patient management.

Reviewer’s Response: Satisfactory reply received

Comment

How many patient had their sputum culture status reverted back to positive

Response

Thank you for this question. None had their sputum culture status reverted back to

Positive

Reviewer’s Response: No further action needed

Comment

Interestingly, ~14% of the study cohort was initiated on ATT beyond 30 days of RRTB

diagnosis, was not found to affect the outcome. You may consider highlighting that

delay in initiation for SLD.

Response

Thank you for pointing this out. We have mentioned the following in the discussion

section: “Last, a significant number of patients delayed initiation of MDR-TB treatment

by more than 30 days after RR-TB diagnosis. Whilst this was not associated with

having an unfavourable outcome, this has got dire consequences at an individual level

towards disease progression and at a population-level towards MDR-TB transmission

in the community.”

Reviewer’s Response: No further action required

Conclusion: Most of the queries are answered with satisfactory responses. However, the data collection is deficient in terms of several key indicators and weakens the quality of study. This is an inherent weakness of a retrospective study which derives data from NTP. I do appreciate the hard work of the authors in the resource limited setting and understand the difficulties faced. The study can be accepted with minor correction.

Reviewer #3: 1. A language editor is strongly recommended. Acceptance of this manuscript should at the very least be contingent on this.

2. Have the authors looked into the rates and risk factors of unfavorable outcomes separately? i.e. LTFU vs. all-cause mortality vs. unreported etc.? Risk factors may differ across these outcomes and a stratified analysis may add more granularity to this paper (authors should consider at least having these separated outcomes in supplementary tables).

3. Would the ‘not evaluated’ outcome include those patients who were still alive and in care by 24 months who had not completed treatment? These patients could be different to those who have met the definition for LTFU, which speaks to the point above.

4. Gender breakdown listed in abstract is different from Results section. Keeping the reference category the same may make the manuscript easier to read.

5. Missing >10% of treatment doses seems to be ascertained throughout treatment and not a once off measurement at baseline? If this is the case, then this cannot be used as a predictor of unfavorable outcome. “Percentage of missed doses: percentage of the number of days with missed doses divided by the total number of days a patient was on treatment up until date of outcome.”

6. “study findings were reported in accordance with Strengthening the Reporting of Observational Studies in Epidemiology (STROBE) guidelines.” This should be included in the methods and not discussion.”

7. PLOS authors have the option to publish the peer review history of their article (what does this mean?). If published, this will include your full peer review and any attached files.

Reviewer #1: No

Reviewer #2: No

Reviewer #3: No

---

## [Author Response · Author response to Decision Letter 1]

29 Jan 2020

Response to Reviewers 

Rebuttal Letter

PLOS ONE

The Editor in Chief

23 December 2019

REF: PONE-D-19-16150: Treatment outcomes of Multi drug resistant and Rifampicin resistant Tuberculosis in Zimbabwe: A cohort analysis of patients initiated on treatment during 2010 to 2015

Dear Editor

Thank you very much for reviewer comments and suggestions during second round review of our above named manuscript. We have carefully read through the reviews and revised our paper accordingly. We feel that the paper is much improved as a result of this review process and thank you for taking it to this stage. We have responded point by point to the comments and suggestions of the reviewers.

We would like to submit the revised version of the manuscript with highlights (R2- track) and the revised and clean version of the manuscript (R2-Clean). The line numbers in the response to reviewer comments corresponds to the line numbers in the revised version with track changes.

Sincerely

R. Matambo (Corresponding Author)

 

Reviewer-1

Comment 1: Consider adding – lack of radiological data in the heading of limitation.

Author Response

Thank you for your suggestion. We have added the statement under study limitations and now the paragraph reads as follows: “Second, there were missing data on key variables which include CDST results, socioeconomic status, WHO clinical staging, CD4 cell count, timing of ART in relation to MDR-TB treatment commencement, nutritional status, MDR-TB drug regimens and their dosages, data on virological, immunological or clinical failure of patients initiated on ART and radiological findings of TB lesions – all which are important factors which may be related to MDR-TB treatment outcomes”. Line 380

Comment 2: Clarification needed for patients – Lost to follow up or died- were the failing on the current regimen.

Author Response

Thank you for raising this important query.

As this was a retrospective study, we don’t have data to safely conclude whether these patients were failing on the current regimen. Of the 164 patients who died or were LTFU, the majority 112 (68.3%) had no culture conversion result recorded. Of the remaining 52 (31.7%), 48 (94.2%) had culture conversion within 6 months of initiating MDR-TB treatment. Thus, we have not tried to interpret these findings. However, we acknowledge this as a deficiency of the current study.

Comment 3: Details of ART initiation is not provided neither the status of ART success or failing is analysed. Please consider mentioning this shortcoming in the Discussion.

Author Response

Thank you for suggesting that we mention in the discussion section about the un-availability of data on virological, immunological or clinical failure of patients initiated on ART. This has now been mentioned in line 370-371

Comment 4: “study findings were reported in accordance with strengthening the Reporting of Observational Studies in Epidemiology (STROBE) guidelines.” This should be included in the methods and not discussion.” 

Author Response

Thank you for such valid suggestion. We have since incorporated the changes as advised. Line 287-288

Comment 5: A language editor is strongly recommended. Acceptance of this manuscript should at the very least be contingent on this.

Author Response: 

Thank you for this suggestion. We have engaged a well experienced scientific editor who has gone through the manuscript and made grammatical changes to the manuscript. The grammatical changes have been made throughout the manuscript. 

 

Reviewer 2:

Comment: Have the authors looked into the rates and risk factors of unfavorable outcomes separately? i.e. LTFU vs. all-cause mortality vs. unreported etc.? Risk factors may differ across these outcomes and a stratified analysis may add more granularity to this paper (authors should consider at least having these separated outcomes in supplementary tables). 

Author Response: 

Thank you for the comment. You raise a very valid point. However, we have another paper in which we had more granular analysis on rates and risk factors of specific unfavorable outcomes (LTFU and death) which if included in this paper would result in duplication. We hope that this is acceptable with the reviewer and the editor if this further analysis is not included in this manuscript.

Comment 2: Would the ‘not evaluated’ outcome include those patients who were still alive and in care by 24 months who had not completed treatment? These patients could /be different to those who have met the definition for LTFU, which speaks to the point above. 

Response: Thank you for the comment. None of the patients in this cohort were still on the treatment at the time of censoring date. Thus, we have not considered anyone still on treatment as ‘not evaluated’. Within the TB programme in Zimbabwe, MDR/RR-TB patients who would have completed their MDR-TB treatment with 1) a negative bacteriological confirmation result are considered cured whilst 2) those without bacteriological confirmation are classified as treatment completed. In response to your question above, only 3/33 (9.1%) were LTFU after being on treatment for 24 months. This implies that they were LTFU before completing their extended treatment duration which was a result of having missed treatment for a duration(s) not exceeding 2 months.

Comment 3: Gender breakdown listed in abstract is different from Results section. Keeping the reference category the same may make the manuscript easier to read.

Author Response: 

We agree with your comment. We have therefore changed the narrative in the results section to match the reference category in the abstract i.e. “….230 (49%) were males”. Line 324-325

Comment 4: Missing >10% of treatment doses seems to be ascertained throughout treatment and not a once off measurement at baseline? If this is the case, then this cannot be used as a predictor of unfavorable outcome. “Percentage of missed doses: percentage of the number of days with missed doses divided by the total number of days a patient was on treatment up until date of outcome.”

Author Response: 

Thank you for highlighting the deficiency of this variable. We acknowledge that this cant be used as a predictor. Hence, we have removed variable on missed doses from our univariate and multivariate regression model. We have also removed narrative in the abstract, results section and discussion section with reference to this predictor. Lines 111-112; 349-350; 362; 406-409; Table 5.

Comment 5: “study findings were reported in accordance with strengthening the Reporting of Observational Studies in Epidemiology (STROBE) guidelines.” This should be included in the methods and not discussion.”

Author Response

Thank you for such valid suggestion. We have since incorporated the changes as advised. Changes made in line number 370-371

We thank you for your esteemed review of our manuscript and hope that we have responded satisfactorily. Please do not hesitate to contact us for further clarification as your reviews enriches our manuscript.

---

## [Decision Letter · Decision Letter 2]

11 Mar 2020

Treatment outcomes of Multi drug resistant and Rifampicin resistant Tuberculosis in Zimbabwe: A cohort analysis of patients initiated on treatment during 2010 to 2015

PONE-D-19-16150R2

Dear Dr. Matambo,

We are pleased to inform you that your manuscript has been judged scientifically suitable for publication and will be formally accepted for publication once it complies with all outstanding technical requirements.

With kind regards,

Denise Evans, PhD

Academic Editor

PLOS ONE

Additional Editor Comments (optional):

Reviewers' comments:

Reviewer's Responses to Questions

**Comments to the Author**

1. If the authors have adequately addressed your comments raised in a previous round of review and you feel that this manuscript is now acceptable for publication, you may indicate that here to bypass the “Comments to the Author” section, enter your conflict of interest statement in the “Confidential to Editor” section, and submit your "Accept" recommendation.

Reviewer #1: All comments have been addressed

Reviewer #2: All comments have been addressed

Reviewer #3: All comments have been addressed

2. Is the manuscript technically sound, and do the data support the conclusions?

Reviewer #1: Yes

Reviewer #2: Yes

Reviewer #3: Yes

3. Has the statistical analysis been performed appropriately and rigorously? 

Reviewer #1: N/A

Reviewer #2: Yes

Reviewer #3: Yes

4. Have the authors made all data underlying the findings in their manuscript fully available?

Reviewer #1: Yes

Reviewer #2: Yes

Reviewer #3: Yes

5. Is the manuscript presented in an intelligible fashion and written in standard English?

Reviewer #1: Yes

Reviewer #2: Yes

Reviewer #3: Yes

6. Review Comments to the Author

Reviewer #1: Please check the references list: there is discrepancy between the first revision and the second revision with 2 references lost.

Reviewer #2: Except the fact that key details could not be collected like radiological data, which may have affected the outcome or the quality of study. Otherwise, The study is acceptable.

Reviewer #3: The authors have made a substantial improvement to this manuscript since their previous submission. This is both in terms of the statistical appropriateness and clarity of their presentation. The manuscript now provides a detailed insight into the Zimbabwean NTP and identifies valuable programmatic factors that may contribute to poor outcomes among patients infected with TB. There are a few minor clarifications that are needed and authors should pay special attention to punctuation and typographical errors, particularly at this stage of their submission.

Overall, there has been great improvment and the authors should be commended on their efforts.

7. PLOS authors have the option to publish the peer review history of their article (what does this mean?). If published, this will include your full peer review and any attached files.

Reviewer #1: No

Reviewer #2: Yes: Prashant Chhajed

Reviewer #3: No

---

## [Editor Report · Acceptance letter]

16 Mar 2020

PONE-D-19-16150R2 

Treatment outcomes of Multi drug resistant and Rifampicin resistant Tuberculosis in Zimbabwe: A cohort analysis of patients initiated on treatment during 2010 to 2015 

Dear Dr. Matambo:

I am pleased to inform you that your manuscript has been deemed suitable for publication in PLOS ONE. Congratulations! Your manuscript is now with our production department. 

With kind regards,

on behalf of

Dr. Denise Evans 

Academic Editor

PLOS ONE